# Anaerobic bacteria *Cetobacterium* sp. nov C33 plays a crucial role in the intestinal microbial balance and regulation of gene expression to immune and metabolic responses in Nile tilapia

**Mario Andrés Colorado Gómez** [ID][1], **Javier Fernando Melo-Bolívar**[1], **Ruth Yolanda Ruíz Pardo**[1], **Howard Junca** [ID][2], **Juan F. Alzate**[3], **Luisa Marcela Villamil Díaz** [ID][1]*

**1** PhD Program in Biosciences, Universidad de La Sabana, Chia, Colombia, **2** RG Microbial Ecology: Metabolism, Genomics & Evolution, Div. Ecogenomics & Holobionts, Microbiomas Foundation, Chía, Colombia, **3** Centro Nacional de Secuenciación Genómica- CNSG, Sede de Investigación Universitaria SIU, Facultad de Medicina Universidad de Antioquia, Medellín, Colombia

* luisa.villamil@unisabana.edu.co

## Abstract

Aquaculture ranks among the largest global food production industries, with Nile tilapia (*Oreochromis niloticus*) being one of the most widely farmed species. However, increasing consumer demand and higher stocking densities place considerable stress on aquaculture systems, often leading to a rise in fish diseases. Probiotics have emerged as valuable tools in this sector, promoting fish health by modulating physiological functions such as metabolism, digestion, immune responses, stress tolerance, and disease resistance. Here, the probiotic potential of *Cetobacterium* sp. nov C33, an anaerobic bacterium isolated from the intestine of Nile tilapia, was evaluated on short-term dietary supplementation in fingerlings in laboratory conditions. Using 16S rRNA amplicon sequencing, we assessed the impact of *Cetobacterium* sp. nov C33 on gut microbiota, while transcriptomic analysis of the head kidney provided insights into immune system modulation. Results indicate that dietary inclusion of this anaerobic bacterium significantly alters the gut microbiota structure in tilapia fingerlings and regulates genes associated with key metabolic pathways, including the immune system, underscoring its potential as a probiotic for enhancing tilapia health. Our results offer promising evidence of its potential as a probiotic to improve tilapia health.

## Introduction

Nile tilapia is an omnivorous freshwater fish that has become one of the world's predominant farmed species with an annual production of over 4.5 million tons [1]. However, culture conditions and environmental factors, such as poor water quality,

**Data availability statement:** Data are publicly available at Zenodo repository (DOI): 10.5281/zenodo.17479128. Sequence datasets have been deposited in the NCBI Sequence Read Archive (SRA) as follows: 16S rRNA gene amplicons of fish gut microbiomes (Metagenome): BioProject PRJNA1306746, BioSample SAMN50644908 and Fish head kidney transcriptomic datasets (Transcriptome): BioProject PRJNA1306769, BioSample SAMN50645431.

**Funding:** This research was funded by the Sistema Nacional de Regalías (SGR): Ministerio de Hacienda de Colombia y Ministerio de Ciencias y Tecnología e Innovación de Colombia. Project code before the National Bank of Investment Programs and Projects bpin: 2020000100487. The project was carried out by the Universidad de La Sabana and the Servicio Nacional de Aprendizaje -SENA-Regional La Guajira. The funders had no role in study design, data collection and analysis, decision to publish, or preparation of the manuscript.

**Competing interests:** The authors have declared that no competing interests exist.

high stocking density, deficient nutrition, temperature changes, handling, transportation, and disease treatment, are making them more susceptible to infections caused by emerging and re-emerging pathogens, such as *Streptococcus agalactiae* Ia and Ib [2], tilapia lake virus (TiLV) [3], and infectious spleen necrosis virus (ISKNV) infections [4], associated to a mass mortalities in tilapia farms.

It is known that fish microbiota is associated with fish health, and it can be affected by the culture system and rearing water, as well as the establishment of gut microbiota in early life stages [5]. Previous research has indicated that the intestinal bacterial composition of tilapia is primarily composed of three phyla: Fusobacteria, Bacteroidota, and Proteobacteria [6,7]. Among the most common bacteria in tilapia are *Cetobacterium*, *Lactobacillus*, *Legionella*, *Lactococcus*, *Rhodobacter*, *Pelomonas*, and *Streptococcus* [8].

Probiotics are defined as microorganisms that confer health benefits to the targeted host, among those traits, maintaining a healthy balance between pathogenic and non-pathogenic bacteria [9]. The study of the microbiota and the use of probiotic sources has been mainly focused on aerobic bacteria [10], however, anaerobic bacteria like *Cetobacterium* have been found as a predominant genus in the microbiota of the intestines of many marine and freshwater fish [11–13] furthermore, *Cetobacterium* produces metabolites which could contribute to the positive effect on the intestinal epithelial barrier, the intestinal health, liver health, altered gut microbiota composition on fish and also protect fish from spring viraemia of carp virus (SVCV) infection [14–16].

Recently, Zhang *et al*., [17] isolated *Cetobacterium* from the intestine of Nile tilapia and analyzed its draft genome and *in vitro* function, finding that *Cetobacterium* was the predominant genus in the foregut, midgut, and hindgut of tilapia. They identified genes related to amino acid and carbohydrate metabolism and reported various functions, including the production of amino acids and vitamins, suggesting that *Cetobacterium* plays a crucial role in fish health.

Colorado *et al*., [18] isolated *Cetobacterium* sp. nov C33 from the intestinal microbial content of Nile tilapia under anaerobic conditions. The phylogenetic analysis suggests that the C33 strain represents a new candidate species of the genus *Cetobacterium*. C33 showed adaptability to the host gastrointestinal conditions, antibacterial activity against aquaculture bacterial pathogens, and antibiotic susceptibility. Furthermore, the whole genome was sequenced, annotated and subjected to functional inference, particularly regarding reported probiotic activities. This analysis highlighting that *Cetobacterium* sp. nov C33 possesses a set of genes encoding sactipeptides, the bacteriocins Linocin M18 and Zoocin A, which may contribute to antagonism against enteric pathogens [19] such as *Streptococcus agalactiae* and *Aeromonas hydrophila*. Also, *Cetobacterium* sp. nov C33 can survive even at pH 2.0 and bile salts, has positive esterase activity, is a gamma (γ) hemolytic bacterium, with hydrophobicity percentages higher than 50% with a non-polar solvent (chloroform) and can biosynthesize amino acids. However, the functions of *Cetobacterium* in the fish intestine require further investigation through *in vivo* and *in vitro* experiments [17,18,20,21].

The purpose of this work was to evaluate the *in vivo* effect of the short-term (5 days) feed administration of the anaerobic bacteria *Cetobacterium* sp. nov C33 on the modulation of the intestinal microbiota and the immune system of Nile tilapia fingerlings (*O. niloticus*) as a strategy to preserve the microbial balance, to prevent dysbiosis of tilapia in aquaculture and to increase sustainable tilapia productivity through the search for new potential probiotics.

## Materials and methods

### Ethical statement

The ethics review board approved the animal care and experimental protocols in accordance with Universidad de La Sabana N° 57 of 2016, and the international ethical guidelines for animal experiments were followed in accordance with Directive 2010/63/EU. Furthermore, following Colombian national government regulations, "Permits for accessing genetic resources were issued by the Colombian Ministry of Environment Number 117, 26 of May 2015 and Otrosí 4 of 2018".

The experiments involved animals, therefore, the methods and protocols were carried out according to ARRIVE guidelines and regulations [22] and all methods were performed in accordance with the relevant guidelines and regulations of the Management and Use of Laboratory Animals of Colombia and complied with Colombia's existing laws and regulations for biological research. Furthermore, the euthanasia protocol sought to reduce fish pain in the experiment, implementing euthanasia as an animal welfare method. Therefore, an overdose of tricaine methanesulfonate (MS-222, 200 mg L-1) [9] was administered by prolonged immersion. Fish were submerged in a tricaine solution for more than 5 minutes or until no opercular movement was observed. Likewise, for the efforts to alleviate suffering, two of the 3R criteria were taken into account: reduction, which refers to using the smallest possible number of animals to obtain a meaningful and valid result [23]. Refinement, which is related to improving handling techniques and reducing pain and suffering in research animals, was also considered. Welfare criteria were established for the fish, such as morphological characteristics (loss of scales, body lesions, etc.) and behavioral characteristics that could be caused by the experimental handling of the animals, which are under stress in certain situations.

### Bacterial strain

*Cetabacterium* sp. nov C33 was isolated from Nile tilapia gut microbiota [18]. Genome sequence data *Cetobacterium* sp. nov C33 has been deposited under GenBank accession number JAVIKH000000000. Source samples were the intestinal content of cultured Nile tilapia. Serial dilutions were made in phosphate buffer (pH 7.3) containing 0.05% hydrochlorinated L-cysteine and 0.001% resazurin under anaerobic conditions [24]. Then, 100 µL were plated on Columbia agar culture medium at pH 7.22 with 5% lamb red blood cells incubated overnight at anaerobic conditions ($O_2$: below 1%; $CO_2$: 9–13%; 28 °C) in an anaerobic jar (2.5 L AnaeroJar, Oxoid, Hampshire, England) [6,18]. Strain *Cetobacterium* sp. nov C33 was cultured anaerobically in an anaerobic jar in Columbia agar at 28 °C for 24 hours until reaching a concentration of $10^8$ CFU/mL. The isolate was stored at −20 °C in Columbia Broth culture medium containing 20% glycerol [9]. In addition, bacterial cultures inocula were prepared by incubation in Columbia Broth containing 0.05% hydrochlorinated L-cysteine and 0.001% resazurin under anaerobic conditions ($O_2$: below 1%; $CO_2$: 9–13%) at 28°C overnight.

### Diets preparation

**Freeze drying and storage.** Anaerobic bacteria *Cetobacterium* sp. nov C33 was included in the balanced feed at a concentration of $10^8$ cells/g feed [15,25]. The composition of the feed was 45% crude protein, 4% crude lipids, 5% fiber, 10% ash and 12% moisture (Agrinal Tilapia 45, Solla, Itagüí, Colombia). Due to the anaerobic nature of the bacteria used in this study, the bacteria and feed were placed in 15 ml Falcon tubes, maintaining the anaerobic environment by continuously supplying 99% $CO_2$ until each tube was tightly sealed with the daily fish ration. This mixture was lyophilized (Lyophilizer LABCONCO 64132, MISSOURI E.E.U.U) and vacuum-packed. Batches of mixture (feed and probiotic) were

used daily for the five days of experimentation. Samples of preserved feed (n = 3 replicates) were taken at 24 hours, 48 hours and 72 hours after starting the experiment to define the decrease in CFU/g of *Cetobacterium* sp. nov C33 during storage. Briefly, 1 g of each stored sample was resuspended in 10 mL TBS under anaerobic conditions and 100 μL was taken and inoculated onto Columbia agar plates at pH 7.22 and incubated under anaerobic conditions ($O_2$: below 1%; $CO_2$: 9–13%; 28 °C) in an anaerobic jar (2.5 L AnaeroJar, Oxoid, Hampshire, England) at 28 °C for 48 h to determine CFU/g of bacteria. Cell counts were Log10 transformed for comparisons. All data were expressed as mean ± standard error of the mean (SEM) [9].

## Experimental design

Male tilapia fingerlings were donated by a commercial fish farm from Meta, Colombia, and acclimatized to experimental conditions for two weeks. The alevins of Nile tilapia (1.05 ± 0.08 g) were randomly distributed into 20-L aquaria at 5 fish per aquarium with four replicate tanks per experimental treatment; commercial feed diet + *Cetobacterium* sp. nov C33 -C33D- and control Diet -CD- (commercial feed). Water quality parameters were monitored daily throughout the acclimation and the feeding trial. Temperature, dissolved oxygen, and pH were recorded using a multiparameter sonde (Hanna Instruments HI9829). Water quality was maintained within ranges suitable for Nile tilapia, water temperature 28 °C, dissolved oxygen above 5 mg L$^{-1}$, oxygen saturation 85%, and pH 7.12.

## Modulation of intestine microbiota in Nile tilapia fingerling

**Data analysis of intestinal microbiota sequences.** The composition of gut-associated microbiota was analyzed using culture-independent 16S rRNA gene sequencing via Illumina technology. At the end of the experiment, after 5 days of treatment, the tilapia intestines were sampled aseptically from the experimental groups, with three replicates, before fish exposure to an overdose of tricaine methanesulfonate (MS-222, 200 mg L-1) [9]. Microbial genomic DNA was isolated from each intestine using a Stool DNA Isolation kit (Norgen Biotek Corp, Thorold, ON, Canada) according to the manufacturer's instructions. Electrophoresis on 1% agarose gel was used to confirm the DNA quality of the isolated microbial genomic DNA. The PCR, library construction, and NGS sequencing were run using V4 and V5 region of the bacterial 16S rRNA using 5'-barcode tagged primers 515F 5'-GTGCCAGCMGCCGGTAA-3' and 907R 5'-CCGTCAATTCCTTTGAGTTT-3, where each barcode contained an eight-base sequence unique to each sample. The sequencing libraries were prepared according to the manufacturer's guidelines using NEBNext® Ultra™ II DNA PCR-free Library Prep Kit for Illumina®. The library's quantification and validation were determined using a Thermo Scientific Qubit 2.0 Fluorometer and an Agilent Bioanalyzer 2100 machine.

Furthermore, following conventional protocols, the library was sequenced using paired-end reads (2 x 250 bp) on an Illumina NovaSeq 6000 platform. FastQC (version 0.11.5) software was used to assess the quality of Illumina's raw fastq data files. QIIME 2.0 (v.2018.8) pipeline was used to process raw sequence data [26]. Following demultiplexing, the sequences were subjected to quality control using DADA2 in RStudio; raw sequences with an average quality score of less than 30 were discarded; chimeras, singletons, and primers were discarded and removed from the data using the same package [27]. The last step resulted in datasets for each amplicon that were assembled to detect the counts of each unique Amplicon Sequence Variant (ASV) across all samples with 100 percent sequence identity and classified according to the SILVA 138 dataset. Classified ASVs higher taxonomic hierarchies were manually curated according to the most recent Prokaryotic Code by the International Committee on Systematics of Prokaryotes (ICSP) [28].

**Estimation of microbial diversity.** To estimate microbial diversity, abundance tables were normalized with Total Sum Scaling (TSS) in each sample [29,30]. The starting point was a data table where each row represents a sample and each column an OTU (operational taxonomic unit). The values in the table are the read counts of each OTU in each sample. Data filtering and normalization were performed to remove low-quality or uninformative features from the raw abundance data, improving subsequent statistical analysis. Outliers in the abundance data were identified

using interquartile range (IQR)-based filtering, which is robust in handling skewed microbiome data and minimizes the influence of extreme values that can distort the overall analysis. Data filtering and normalization were performed to remove low-quality or uninformative features from the raw abundance data, thereby improving subsequent statistical analysis [31]. Features with very low counts (fewer than 4 reads in less than 20% of samples) were filtered out to exclude rare taxa, focusing the analysis on microbes with consistent presence and biological relevance [32,33]. A low-variance filter, using RIC variances, was applied to remove features with low variability between samples, ensuring robustness in subsequent statistical analyses. Normalization addressed variability in sampling depth and data sparsity, and values were normalized by dividing the number of reads for each OTU by the total number of reads in each sample, converting counts to proportions [34]. Each value in the table represents the proportion of reads for a specific OTU relative to the total reads in the sample. Once the data were normalized with TSS, Alpha diversity indices were calculated, such as the measure of observed species richness with Chao1 index, and species diversity with ASVs the Shannon index measure [35]. The statistical significance between groups of alpha diversity was calculated using a non-parametric test depending on the number of factors of the variable to be studied: the Wilcoxon-Mann-Whitney test, if the variable had two factors, or the Kruskal-Wallis if the variable presented three or more factors, with the subsequent calculation of the Wilcoxon-Mann-Whitney test for each pair of factors. When making multiple comparisons, all $p$-values were adjusted using the Bonferroni test [36].

A Bray–Curtis dissimilarity matrix was constructed for the bacterial community and functional analyses, including a preliminary one-way permutational multivariate analysis of variance (PERMANOVA) and a principal coordinates analysis (PCoA). PERMANOVA was used to examine the diversity of the bacteria in the different groups ($p < 0.05$ was considered statistically significant) [37]. Microbiome Analyst 2.0 has been used as an analysis and image generating platform [38].

## Modulation of the immune system in Nile tilapia fingerling

After five days of the mentioned treatment administration, Control Diet -CD- (Agrinal Tilapia 45 feed Solla, Itagüí, Colombia) and Experimental Diet -C33D- (Agrinal Tilapia 45 + *Cetobacterium* sp. nov C33), fish were euthanized with an overdose of tricaine methane sulfonate (MS-222) [9]. External disinfection of the fish was performed with 70% ethanol. For RNAseq sequence analysis, the head kidney was removed aseptically and stored in sterile tubes with TRIZOL at −80°C for RNAseq analysis.

The total RNA was extracted from the samples using TRIZOL reagent® (Invitrogen, Carlsbad, CA) according to the manufacturer's protocol. Then, total RNA quantification, quality assessment, rRNA depletion, library preparation, and RNA sequencing were carried out. In summary, total RNA QC was run for sample quantification (Nanodrop and Bioanalyzer 2100, Agilent), sample integrity (Bioanalyzer 2100 (Agilent); chip: Agilent RNA 6000 Nano Reagents Part I), and sample purity (Agarose gel and Bioanalyzer 2100, Agilent). Then, according to the manufacturer's instructions, total RNA was subjected to rRNA-depletion treatment using a Ribo-ZeroTM Magnetic kit (Illumina). An RNA-Seq library was then constructed using a Next Ultra Directional RNA Library Prep Kit (NEB, Ipswich, MA, USA) and sequenced on an Illumina Novaseq Platform (PE150).

Raw reads were trimmed of adapter sequences with CUTADAT 4.6 and mapped to the Nile tilapia genome with accession number GCA_001858045185 (https://www.ebi.ac.uk/ena/browser/view/GCA_001858045.3?show=blobtoolkit) with STAR short read aligner version 2.7.10a (https://github.com/alexdobin/STAR) with default settings. The output read counts per gene were used for the analysis of Differentially Expressed Genes (DEGs) [39]. Differentially expressed genes were obtained with the R package edgeR (version 3.26.5) including the paired nature of the sample with FDR < 0.05 [40]. Based on the functional annotation of the unigenes, all the DEGs in treatments Control Diet -CD-, and *Cetobacterium* Diet -C33D- were submitted to the KofamKOALA web server (https://www.genome.jp/tools/kofamkoala/) for assigning KEGG Orthologs (KOs) from the protein sequences downloaded from NCBI gene dataset (https://www.ncbi.nlm.nih.gov/datasets/tables/genes/) [41]. Then the KOs were submitted to KEGG Mapper to identify the pathway of each gene.

## Statistical analysis

The statistical software R (https://www.R-project.org/, Vienna, Austria), version 4.0.3, was used to process all data. For the statistical analysis, the normality of the data and the homogeneity of variance analysis were determined using the Shapiro-Wilk and Levene test, respectively. Data are presented as mean values of standard error mean (SEM). Data were analyzed using one-way ANOVA with a significance set at $p \leq 0.05$. Duncan's multiple range test was conducted to compare the variations between treatment means a significant interaction. In addition, orthogonal contrasts were used to compare combinations of treatments; Control Diet -CD- vs. *Cetobacterium* Diet -C33D- [9].

## Results

*Cetobacterium* sp. nov strain C33, originally isolated from the intestinal content of Nile tilapia, was cultured as previously described [18] and from the same bacterial culture the probiotic mixture was prepared, as previously described, for the five days of experimentation, thus maintaining the same concentration in each sample of approximately $10^8$ CFU/g at beginning of the experiment. We found that the concentration on average of viable bacteria 24 hours after starting the experiment was $1.62 \times 10^7$ CFU/mL; at 48 hours it was $8.70 \times 10^6$ CFU/mL and, at 72 hours it was $7.00 \times 10^6$ CFU/mL.

### Modulation of intestine microbiota in Nile tilapia fingerling

Bioinformatic analysis showed an average of 51,753 validated nucleotide sequences retrieved from the two experimental groups: control diet (CD) and *Cetobacterium* sp. nov C33 supplemented diet (C33D) (S1 File, S2 File, S3 File, S4 File S5 File). The validated nucleotide sequences were clustered into operational taxonomic units (OTUs) with 97% similarity level, which acquired 906 OTUs that were assigned to 18 different phyla. The sample rarefaction curve tended to approach the plateau, and coverage reached 99.9%. The PCoA of the microbiota in the CD and C33D clustered into two distinct groups. The PERMANOVA *p*-value: 0.031 was considered a statistically significant diversity of the bacteria between two groups.

The α-diversity was assessed using indices such as the Chao1 index and the Shannon index. The Chao1 index was used to calculate richness, which refers to the total number of different species, or OTUs, in a community, in this case in the gut microbiota. On the other hand, the Shannon index was used, which accounts for both species richness and evenness of species distribution in the gut microbiota of tilapia fingerlings [42]. It was found that C33 supplementation in the diet has a significant effect on the richness of intestinal microbiota in Nile tilapia, as measured by the Chao1 index (Fig 1A). However, the C33D diet did not significantly affect intestinal microbiota diversity compared to the control diet CD (Fig 1B).

The composition and relative abundance of intestinal microbiota in tilapia intestine were observed at the phylum level in Fig 2A. The results showed that in the C33D diet-treated fish, the composition of the intestinal microbiota varied considerably compared to control fish. At the phylum level, the dominant phyla in the intestinal bacterial community were Proteobacteria, Fusobacteria, Planctomycetota, and Actinobacteriota (Fig 2B). However, the percentage of the abundance of Proteobacteria, Fusobacteria, and Planctomycetota were different among the two groups C33D and CD (Fig 2A). The relative abundance of Proteobacteria was reduced in C33D group (65,91%) in comparison to the control CD (82,42%). However, Fusobacteriota had the opposite trend, only representing 3,02% in CD, and 15,54% in C33D (Fig 2A). Likewise, the same behavior in relative abundance was observed for Planctomycetota, 4,11% in CD, and 9,71% in C33D.

The administration of *Cetobacterium* sp. nov C33 in the feed caused an overall increase in Class Fusobacteria, Planctomycetes, Chlamydiae, Phycispareae, and Armatimonadia (Fig 2B). On the other hand, Gammaproteobacteria and Cyanobacteria decreased in the C33D group (Fig 2B). Finally, the composition and relative abundance of intestinal microbiota in *O. niloticus* were shown at the genus level in Fig 2C. with the greatest increase observed when exposed to C33D in *Xanthobacter*, *Cetobacterium* y *Tundrisphaera*, and the main reduction of *Burkholderia*, *Allorhizobium*, and *Delftia*.

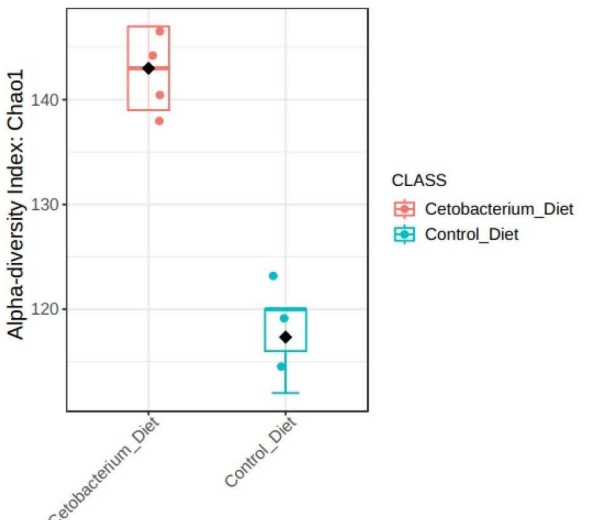
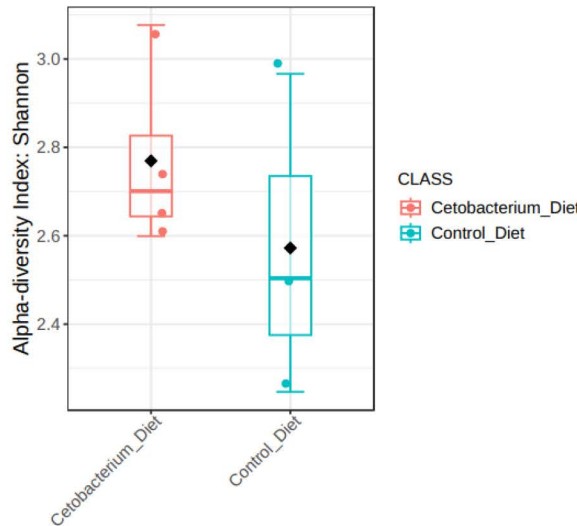

A) Species richness Chao 1 index

B) Species diversity Shannon index

**Fig. 1. Alpha diversity of tilapia gut microbiota was estimated as Chao1 and Shannon indexes.** Alpha diversity analyses were estimated at the feature level as Chao1 and Shannon indexes to analyze the complexity of species diversity in the tilapia gut samples of tilapia fingerlings from the group fed with the *Cetobacterium* diet (C33D): basal diet with 1 x 10^8 CFU/g of *Cetobacterium* sp. nov C33, compared to the control group fed with Control Diet (CD): basal diet; after ceasing consumption of *Cetobacterium* Diet for 5 days. Alpha Diversity Index: (A) Species richness Chao1 index (*p*-value: 0.047678; Mann-Whitney statistic: 12), (B) Species diversity Shannon index (*p*-value: 0.4; Mann-Whitney statistic: 9).

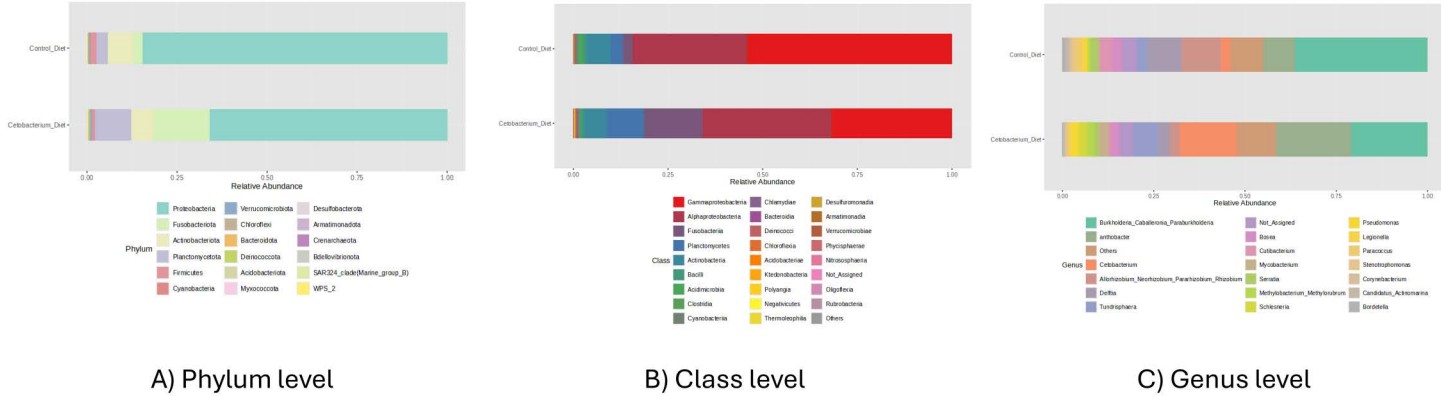

A) Phylum level · B) Class level · C) Genus level

**Fig. 2. Analysis of the compositions of microbes in gut samples of tilapia.** General view using a stacked plot of the taxonomic composition in gut samples of tilapia fingerlings from the group fed with the *Cetobacterium* diet (C33D): basal diet with 1 x 10^8 CFU/g of *Cetobacterium* sp. nov C33, compared to the control group fed with Control Diet (CD): basal diet; after ceasing consumption of *Cetobacterium* Diet for 5 days. A) Phylum level; B) Class level; and C) Genus level.

Differences in the distribution and relative abundance of intestinal microorganisms among the two groups at genus level, are shown in Fig 2C. The bacterial genera that were reduced in the intestine samples of fish fed with *Cetobacterium* Diet -C33D- were *Burkholderia* (CD: 36,47%; C33D 20,95%), *Delfthia* (CD: 9,39%; C33D: 3,29%), *Bosea* (CD: 2,89%; C33D: 2,26%), *Cutibacterium* (CD: 3,21%; C33D: 0,86%) and *Pseudomonas* (CD 0,47%; C33D: 1,61%). Table 1 reports

**Table 1. Bacterial genera with significant reduction in abundance in the intestinal microbiota of Nile tilapia fingerlings in group C33D.**

| Phylum | Class | Order | Family | Genus | log2FC | LfcSE | p-values | FDR |
|---|---|---|---|---|---|---|---|---|
| Pseudomonadota | Betaproteobacteria | Burkholderiales | Oxalobacteraceae | *Massilia* | −2.1242 | 0.7803 | 0.0065 | 0.0447 |
| | Alphaproteobacteria | Hyphomicrobiales | Rhizobiaceae | *Allorhizobium* | −2.1636 | 0.6278 | 0.0006 | 0.0093 |
| | | Rhodobacterales | Rhodobacteraceae | *Lutimaribacter* | −6.7374 | 2.1176 | 0.0015 | 0.0160 |
| | Betaproteobacteria | Burkholderiales | Burkholderiaceae | *Burkhoderia* | −1.2767 | 0.4091 | 0.0018 | 0.0179 |
| | | | Comamonadaceae | *Delftia* | −1.8709 | 0.4120 | 0.0000 | 0.0002 |
| | | | Alcaligenaceae | *Bordetella* | −2.0563 | 0.4882 | 0.0000 | 0.0006 |
| | Gammaproteobacteria | Pseudomonadales | Moraxellaceae | *Moraxella* | −1.7706 | 0.7061 | 0.0122 | 0.0639 |
| | | Xanthomonadales | Xanthomonadaceae | *Luteimonas* | −6.8964 | 2.1334 | 0.0012 | 0.0160 |
| Actinomycetota | Actinomycetia | Propionibacteriales | Propionibacteriaceae | *Cutibacterium* | −2.2369 | 0.3808 | 0.0000 | 0.0000 |
| | | Micrococcales | Intrasporangiaceae | *Janibacter* | −6.2811 | 2.2374 | 0.0050 | 0.0360 |

Principal bacterial genera that significantly decreased relative abundance ($p<0.05$), in gut samples of tilapia fingerlings from the group fed with the *Cetobacterium* diet (C33D): basal diet with 1 x 10$^8$ CFU/g of *Cetobacterium* sp. nov C33, compared to the control group fed with Control Diet (CD): basal diet; after ceasing consumption of *Cetobacterium* Diet for 5 days.

the bacterial genera that had a significant reduction in abundance after feeding the *Cetobacterium* C33D diet compared to the control fish group -CD-. In contrast, the bacteria that increased their relative abundance were *Xanthobacter* (CD: 8,40%; C33D: 20,44%), *Cetobacterium* (CD: 2,61%; C33D: 15,55%), *Thundisphaera* (CD: 2,49%; C33D: 6,62%) and *Mycobacterium* (CD: 0,61%; C33D: 2,62%).

Likewise, Table 2 reports the bacterial genera that had a significant increase in abundance after feeding the *Cetobacterium* C33D diet compared to the control fish group. Besides, in Fig. 3Aa, at the taxonomic level of phylum, the increase in prevalence of Fusobacteriota and Planctomycetota and the reduction in prevalence of Firmicutes can be observed in the gut microbiota samples of tilapia fingerlings that were fed with *Cetobacterium* Diet -C33D- compared to the control

**Table 2. Bacterial genera with a significant increase in abundance in the intestinal microbiota of Nile tilapia fingerlings in group C33D.**

| Phylum | Class | Order | Family | Genus | log2FC | lfcSE | p-values | FDR |
|---|---|---|---|---|---|---|---|---|
| Fusobacteriota | Fusobacteriia | Fusobacteriales | Fusobacteriaceae | *Cetobacterium* | 1.9229 | 0.7348 | 0.0089 | 0.0512 |
| Bacillota | Bacilli | Lactobacillales | Lactobacillaceae | *Lactobacillus* | 5.7593 | 2.3620 | 0.0148 | 0.0714 |
| | | Bacillales | Bacillaceae | *Bacillus* | 7.6980 | 1.7026 | 0.0000 | 0.0002 |
| Pseudomonadota | Alphaproteobacteria | Rhodobacterales | Rhodobacteraceae | *Rhodobacter* | 2.8416 | 1.4266 | 0.0464 | 0.1483 |
| | | Sphingomonadales | Sphingomonadaceae | *Sphingomonas* | 1.9682 | 0.6622 | 0.0030 | 0.0278 |
| | | Hyphomicrobiales | Methylobacteriaceae | *Methylobacterium* | 1.6883 | 0.6620 | 0.0108 | 0.0602 |
| | | | Xanthobacteraceae | *Ancylobacter* | 1.5086 | 0.6843 | 0.0275 | 0.1048 |
| | | | Methylocystaceae | *Methylocystis* | 6.2168 | 2.1868 | 0.0045 | 0.0358 |
| | Gammaproteobacteria | Xanthomonadales | Xanthomonadaceae | *Xanthomonas* | 6.5609 | 1.9583 | 0.0008 | 0.0120 |
| | Betaproteobacteria | Burkholderiales | Comamonadaceae | *Comamonas* | 1.2491 | 0.5663 | 0.0274 | 0.1048 |
| | | | | *Leptothrix* | 10.0320 | 1.3601 | 0.0000 | 0.0000 |
| Actinomycetota | Actinomycetia | Mycobacteriales | Mycobacteriaceae | *Mycobacterium* | 1.6608 | 0.4619 | 0.0003 | 0.0072 |
| Acidobacteriota | Acidobacteriia | Acidobacteriales | Acidobacteriaceae | *Acidipila* | 8.7111 | 1.7178 | 0.0000 | 0.0000 |

Principal bacterial genera that significantly increased relative abundance ($p<0.05$) in gut samples of tilapia fingerlings from the group fed with the *Cetobacterium* diet (C33D): basal diet with 1 x 10$^8$ CFU/g of *Cetobacterium* sp. nov C33, compared to the control group fed with Control Diet (CD): basal diet; after ceasing consumption of *Cetobacterium* Diet for 5 days.

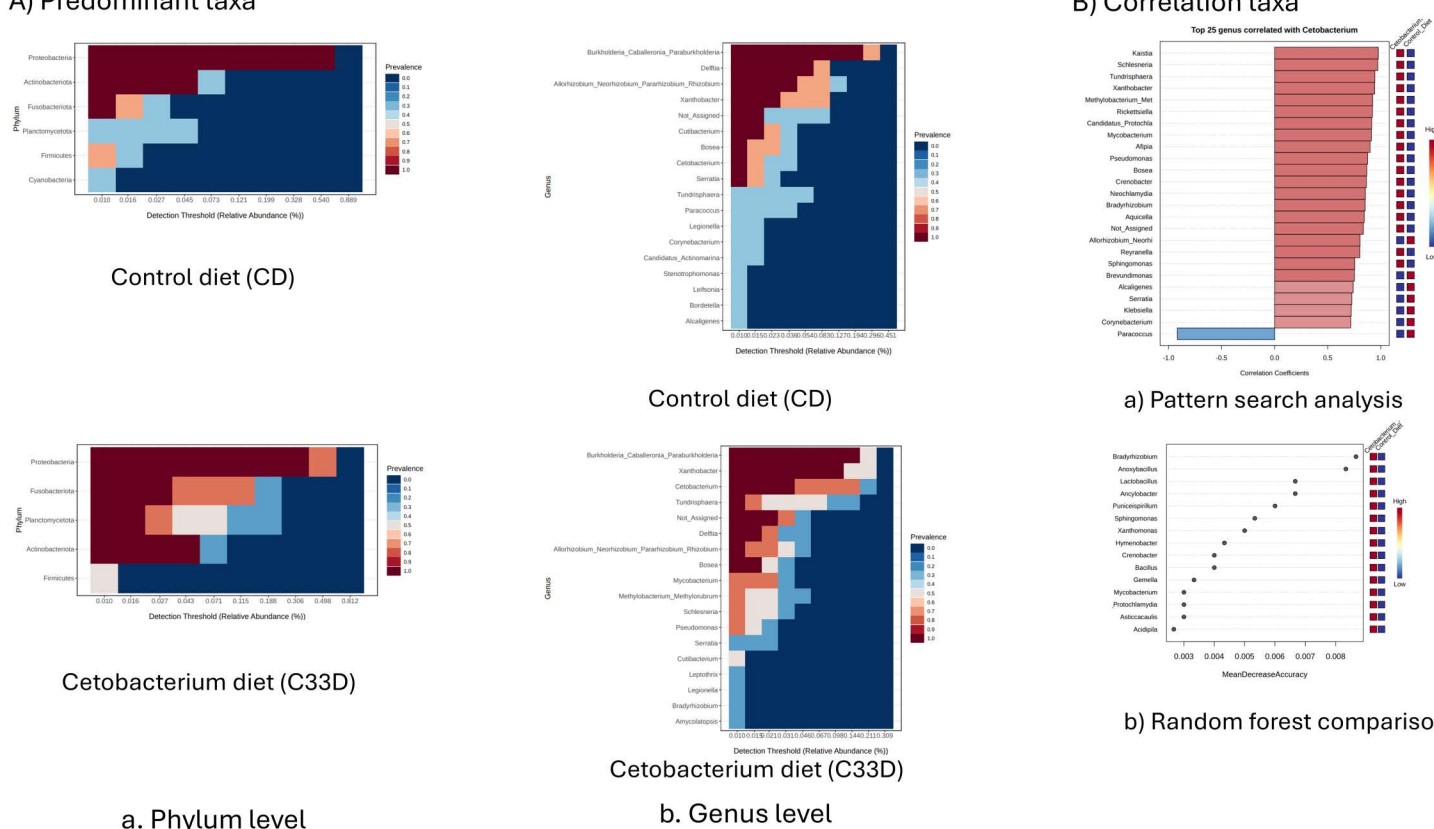

**Fig. 3. Analysis of the correlation of microbes in gut samples of tilapia.** A. Heatmap of predominant taxa among all gut samples. *Cetobacterium* Diet (C33D): basal diet with 1 x10⁸ CFU/g of *Cetobacterium* sp. nov C33; Control Diet (CD): basal diet; after cessation of *Cetobacterium* Diet consumption for 5 days. a) Phylum level; b) Genus level. B. a) Pattern search analysis to identify genera that correlates with *Cetobacterium* in the group of fish fed *Cetobacterium* sp. nov C33. Genera that strongly correlate (coefficients >0.7; positive correlation-red; negative correlation-blue) with the *Cetobacterium* genus in the C33D group are indicated; b) Comparison of Random Forest Model on taxa composition.

samples -CD-. Likewise, in Fig. 3Ab, the increase in prevalence of the genera *Xanthobacter* and *Cetobacterium* observed, and the reduction in the prevalence of *Allorhizobium* and *Delftia*.

It is essential to note that the inclusion of *Cetobacterium* sp. nov C33 in the diet resulted in a higher relative abundance of the *Cetobacterium* genus in the intestinal microbiota, confirming the autochthonous nature and viability of the strain which exhibited major colonization capabilities, permanence, and adaptation to the Tilapia intestine. The major abundance of *Cetobacterium* had a positive correlation with the genus *Kastia*, *Schlesneria, Tundrisphaera, Xanthobacter, Methylobacterium,* and *Rickettsiella*. In contrast, that major abundance had a negative correlation with the genus *Paracoccus* (Fig. 3Ba). In addition, the comparison of the Random Forest Model, in terms of variable importance, revelated that intestinal bacterium, including *Bradyrhizobium, Anoxabacillus, Lactobacillus*, and *Puniceispirillum* were most associated with the C33D group (Fig 3Bb).

**Modulation of the immune system in Nile tilapia fingerling analysis of Differentially Expressed Genes (DEGs)**

A total of 3630 genes from C33 supplemented diet (C33D) and control Diet (CD) groups were identified with clear annotations in the head kidney (S6 File, S7 File, S8 File). The number of DEGs in the CD group was 2,874, and the experimental group C33D was 3.432. Through functional annotation and screening, a total of 327 differentially expressed genes that

were significantly regulated were identified, comprising 50 upregulated DEGs (S1 Table) and 277 downregulated DEGs (S2 Table).

Differentially expressed transcripts in biological processes are represented in Fig 4A showing the main gene expression pathways modulated by C33D treatment, which include metabolism and regulation of cellular processes, growth and development processes, and responses to abiotic stimuli and stress.

Fig 4B illustrates the modulation of gene expression when the diet was supplemented with *Cetobacterium* sp. nov C33 (C33D) in various pathways including the immune system, cellular responses to stimuli, and metabolism, as well as protein metabolism among others.

The scatter plot (Fig 5) shows statistical significance ($p < 0.05$) as a function of fold change magnitude allowing for rapid visual identification of genes with large fold changes that are also statistically significant and where immune response genes stand out. Upregulated immune pathway genes were leucine-rich repeat and immunoglobulin-like domain-containing nogo receptor-interacting protein 3; cystatin-B; beta-galactoside-binding lectin. Gen code down regulated in immune pathway: E3 ubiquitin/ISG15 ligase TRIM25; protein C2-Domain ABA-Related 3-like.

Likewise, the Fig 6 shows the associations between sample and function, where highlighting the up regulated of genes in the immune pathway, such as cystatin-B, transcription factor BTF3 homolog 4, beta-galactoside-binding lectin, leucine-rich repeat and immunoglobulin-like domain-containing nogo receptor-interacting protein 3, LINGO3 gene, PYD domains-containing protein 3-like, class I histocompatibility antigen, F10 alpha chain, C-reactive protein; apoptotic pathway: Cystatin B; tight junction: claudin-8-like gene. In addition, Fig 6 shows the downregulation in gene expression, mainly identifying the reduction in the expression of immune pathway genes found such as interferon GTPase 1-like,

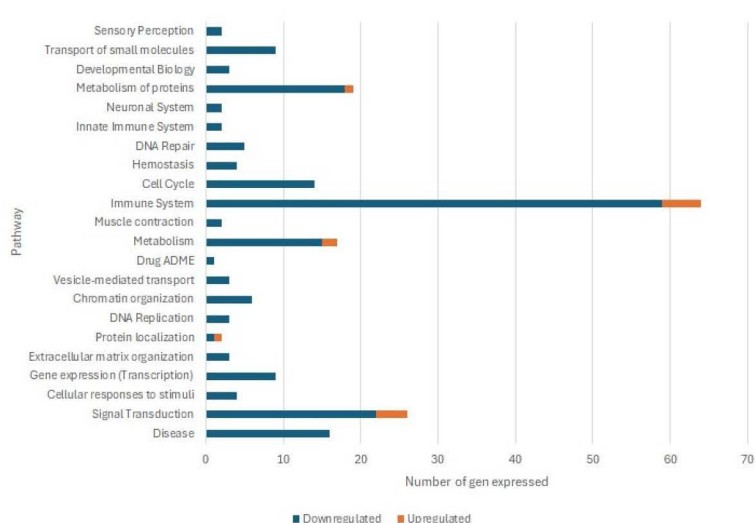

**Fig. 4. Analysis of Differentially Expressed Genes (DEGs).** A. Summary of differentially expressed ESTs using BLAST2GO in C33D in comparison to CD treatments. Transcripts were classified into 6 main GO annotations: Cellular Processes, Diseases, Environmental Information Processes, Genetic information Processes, Metabolism and Organismal Systems. Differentially expressed transcripts in kidney samples of tilapia fingerlings from the group fed with the *Cetobacterium* diet (C33D): basal diet with $1 \times 10^8$ CFU/g of *Cetobacterium* sp. nov C33, compared to the control group fed with Control Diet (CD): basal diet; after ceasing consumption of *Cetobacterium* Diet for 5 days. B. Summary of differentially expressed ESTs using BLAST2GO classified in pathway differentially expressed transcripts in kidney samples of tilapia fingerlings from the group fed with the *Cetobacterium* diet C33D: basal diet with $1 \times 10^8$ CFU/g of *Cetobacterium* sp. nov C33, compared to the control group fed with Control Diet (CD): basal diet; after ceasing consumption of *Cetobacterium* Diet for 5 days.

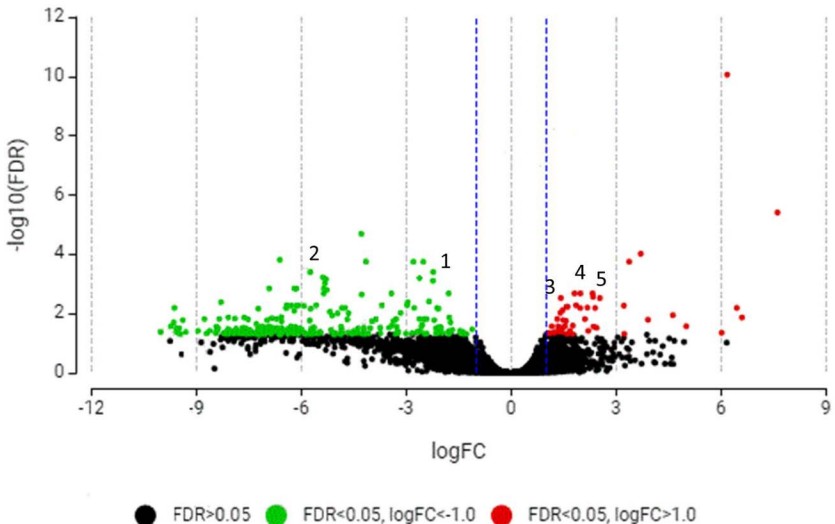

**Fig. 5. Volcano plot of Nile tilapia fingerlings DEGs identified from the *Cetobacterium* sp. nov C33 dietary administration.** C33D vs CD. The log2(FPKM) value represented the mean expression level of each gene, and each dot represented one gene. Differentially expressed transcripts in kidney samples of tilapia fingerlings from the group fed with the *Cetobacterium* diet (C33D): basal diet with 1 x 10^8 CFU/g of *Cetobacterium* sp. nov C33, compared to the control group fed with Control Diet (CD): basal diet; after consumption of *Cetobacterium* Diet for 5 days. The down-regulated genes were shown in green dots, the upregulated genes were shown in red dots and genes with no differential expression were shown in black dots. Gen code down regulated in immune pathway: 1) E3 ubiquitin/ISG15 ligase TRIM25; 2) protein C2-Domain ABA-Related 3-like. Gen code up regulated in immune pathway: 3) leucine-rich repeat and immunoglobulin-like domain-containing nogo receptor-interacting protein 3; 4) cystatin-B; 5) beta-galactoside-binding lectin.

endoplasmatic reticulum protein 27, protein C2-domain aba-related 3-like, macrophage mannose receptor 1-like, endonuclease domain-contaning 1 protein, E3 ubiquitin/ISG15 ligase TRIM25, Perforin. Also, S2 Table shows downregulation in Complement pathway such as complement C1q-like protein 3, complement C1r subcomponent, complement C1r subcomponent-like, complement C2, complement C3, complement C4, complement C8 alpha chain, complement component 8 subunit beta, complement factor H, complement factor H-related protein 1, complement factor H-related protein 5, complement factor I, among other genes.

## Discussion

In the present study, we evaluated *in vivo* the effect of the administration via diet supplementation of the native anaerobic bacterium *Cetobacterium* sp. nov C33 at 10^8 UFC/g feed (C33D), in comparison to the control tilapia diet with no bacteria added (CD). *Cetobacterium* sp. nov C33 is an anaerobic Gram-negative bacterium with negative gelatinase activity, and positive activity of esterase, an enzyme that breaks ester bonds of polysaccharides favoring the action of hydrolases of high molecular weight compounds such as carbohydrates and proteins. In addition to these characteristics, the C33 strain has the capacity to survive even at pH 2.0 and Bile Salt, and has a hydrophobicity percentage higher than 50% [18] indicating the potential ability of this bacterium to colonize and persist in the tilapia gastrointestinal tract, which is now *in vivo* confirmed based on the statistically significant increased detection at the fish intestines under C33 treatment as shown in microbiota analysis.

Despite positive *in vitro* and *in silico* results of the *Cetobacterium* genus, few recent studies evaluate the effect of *Cetobacterium* bacteria *in vivo* experiments. The experiments conducted here, showed a core gut microbiota of Nile tilapia intestines formed by Proteobacteria, Actinobacteriota, Fusobacteria, and Firmicutes as previously reported in different countries, as Colombia [6], Japan [43] and Brazil [13]. Also, we report a modulation of the intestinal microbiota

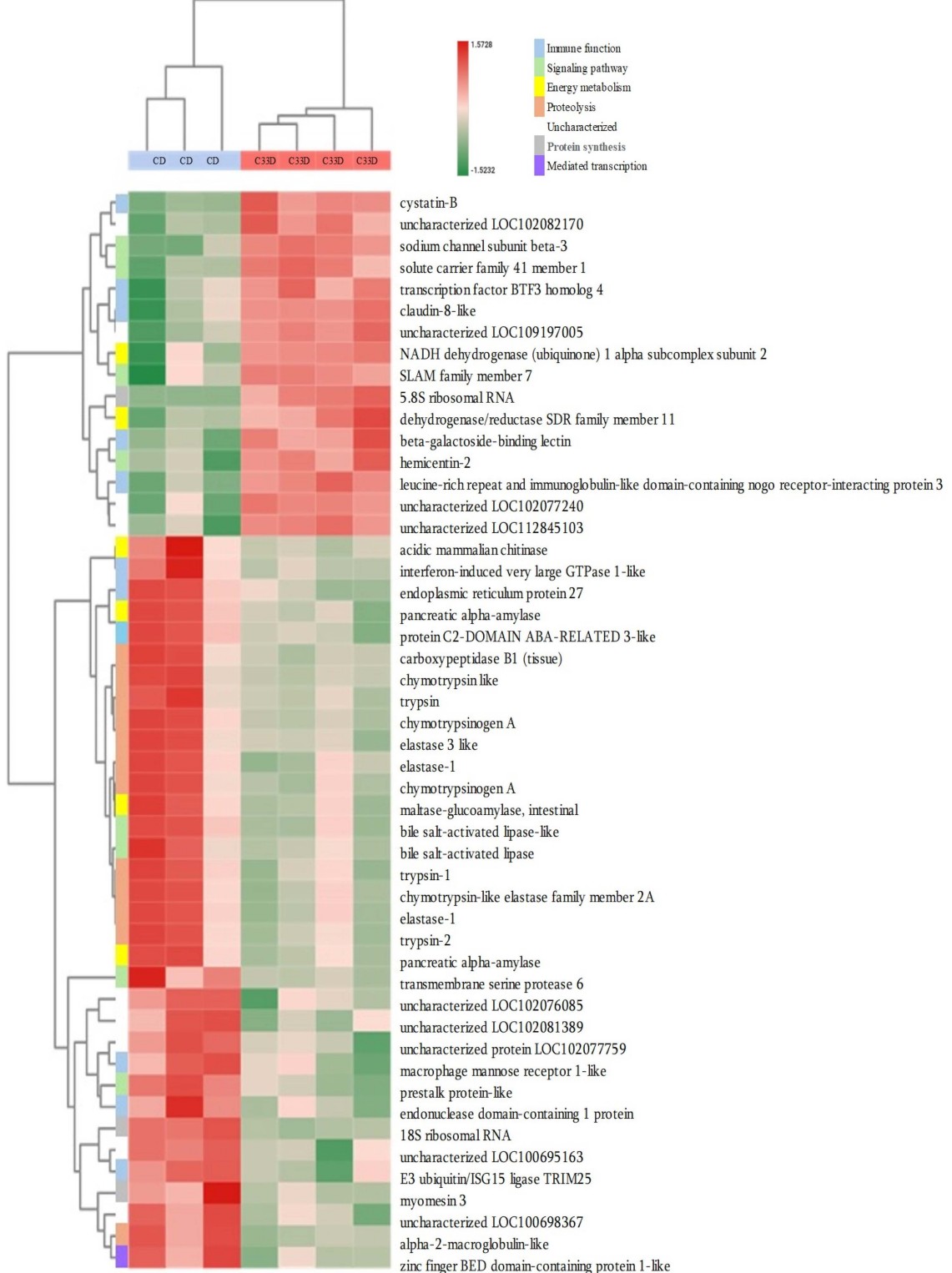

**Fig 6. Associations between sample and function of differentially expressed genes.** Each row corresponds to a gene of each function; each column corresponds to a sample. Differentially expressed genes (DEGs) with significant ($p < 0.05$) difference between kidney samples of tilapia fingerlings from the group fed with the *Cetobacterium* diet (C33D): basal diet with 1 x 10^8 CFU/g of *Cetobacterium* sp. nov C33, compared to the control group fed

with Control Diet (CD): basal diet; after ceasing consumption of *Cetobacterium* diet for 5 days. The upregulated genes were shown in red color, and the down-regulated genes were shown in green color.

composition by the supplementation of the fish diet *Cetobacterium* C33D. The above is similar to what was reported by Zhou *et al.*, [44] who used the stabilized fermentation product of *C. somerae* XMX-1, isolated from the intestine of zebrafish (*Danio rerio*), with a concentration of $3.1 \times 10^8$ CFU/ml, added it to rice husk powder at a ratio of 1:1 (w/w), and dried at room temperature. Their results showed enhanced resistance against tilapia pathogenic bacteria and modulation in the composition of bacteria and viruses in the intestine of tilapia. Furthermore, Xie *et al.*, [15] conducted a study on common carp *Cyprinus carpio* fed a basal diet partially replacing fishmeal with vegetable proteins and supplemented with *C. somerae* fermentation product XMX-1 (uPP + XMX-1), mainly resulting in increased abundance of the phylum Proteobacteria, followed by Firmicutes, Actinobacteriota, Bacteroidota, and Fusobacteria. As well, Li *et al.*, [45] evaluated the effect of N-carbamylglutamate (NCG) on tilapia growth and gut microbiota, to explore the possible mechanism by which NCG promotes tilapia growth from the perspective of gut microbiota, suggesting that dietary supplementation with XMX-1(N) at 100 g/kg improved the gut microbiota of tilapia. Likewise, Xie *et al.*, [20] showed that dietary XMX-1 altered gut microbiota composition in zebrafish. In addition, Hao *et al.*, [46] reported that the abundance of *Cetobacterium* was negatively correlated with Bacteroides, and this balance could be modulated by the diet, with the abundance of *Cetobacterium* bacteria decreasing drastically due to dietary changes in grass carp *Ctenopharyngodon idella*. Our results confirm and expand this observation by directly supplying *Cetobacterium* sp. nov C33 and observing changes in the abundance of phylum such as Pseudomonadota and Proteobacteria, which was reduced in abundance from 82.42% in group CD to 65.91% in C33D group. Within this phylum, the genera that had a significant difference are *Burkholderia* (CD: 36.47%; C33D 20.95%) *Delftia* (CD: 9.39%; C33D: 3.29%), *Xanthobacter* (CD: 8.40%; C33D: 20.44%) *Pseudomonas* (CD 0.47%; C33D: 1.61%). As for the phylum Fusobacteriota, it increased its abundance from 3.02% in CD to 15.54% in the C33D group, with the genus *Cetobacterium* being the largest representative, increasing its abundance (CD: 2.61%; C33D: 15.55%). The above demonstrates the capacity of *Cetobacterium* sp. nov C33 to colonize the gut.

Another phylum that showed a significant difference between the treatments was Planctomycetota with 4.11% in the control group CD and increased to 9.71% in experimental group C33D. Finally, the genera belonging to the phylum Actinomycetota showed variations in abundance between the control group CD and the experimental group C33D; specifically, *Mycobacterium* (CD: 0.61%; C33D: 2.62%), and *Cutibacterium* (CD: 3.21%; C33D: 0.86%). Likewise, the increase in the prevalence of the *Plactomycetota* and *Schlesneira* genera (CD: 0.35%; C33D: 2.17%), which are phylum of Gram-negative, flagellated aquatic bacteria capable of degrading chitin, is highlighted.

Additionally, the treatment with the C33D diet showed a positive incidence in the genera *Lactobacillus*, *Bacillus*, *Anoxybacillus*, and *Effusibacillus* which were previously reported as beneficial microorganisms that colonize the fish gut to stimulate immune responses [21,47–49]. In this regard, *Cetobacterium* sp. nov C33 may be postulated as a commensal bacterium. Furthermore, the treatment group with the C33D diet showed a negative incidence in the relative abundance the bacterial genera involving more potentially pathogenic bacteria, such as *Allorhizobium*, *Burkhoderia*, *Delftia*, *Bordetella*, and *Cutibacterium* [50], the above adds weight to the proposed pivotal modulating role of the microbiota by *Cetobacterium* sp. nov C33.

Commensal microbes are essential in maintaining the development and maturation of the immune system [21]. In recent decades, a variety of gram-positive and gram-negative bacteria have been widely applied as probiotics in aquaculture to regulate immunity and prevent disease outbreaks [51]. Our results demonstrated that dietary administration of *Cetobacterium* sp. nov C33 not only influenced the intestinal microbial balance, but was also able to modulate biological processes, such as the regulation of gene expression. Several biological systems were affected, such as disease response, metabolic processes, and organismal systems.

It is important to note that *Cetobacterium* sp. nov C33 did not activate an inflammatory pathway that could exhaust the fish and compromise its growth capacity or other biological functions. On the contrary, alternative pathways were regulated by the -C33D- treatment in the face of massive inflammation, causing a tendency towards homeostasis and preparing the immune system to act in coordinated and strategic manner.

To the best of our knowledge, this is the first report that studies the immune modulation effects of *Cetobacterium* sp. nov C33 isolated from the intestine of Nile tilapia through transcriptomic analysis of the head kidney, following five days of dietary supplementation in tilapia fingerlings.

Previous studies showed that the use of *C. somerae* XMX-1 isolated from the intestine of zebrafish increased disease resistance in different fish when exposed to experimental infections after supplementation with various bacterial strains and viruses such as spring carp viraemia virus (SVCV), for example in the case of tilapia, Zhou *et al*., [42] in a 7-week feeding trial with a total of 120 tilapia (initial weight $1.33 \pm 0.00$ g), administered the fermentation product of XMX-1 at a concentration of $3.1 \times 10^8$ CFU/ml mixed with rice husk powder at a ratio of 1:1 (w/w), added to the feed according to each diet, obtaining the main results of improving intestinal and liver health, resistance against pathogenic bacteria in tilapia and modulating the composition of bacteria and viruses in the intestine of tilapia, with an enrichment of the beneficial bacterial genus. Furthermore, in carp Xie *et al*., [16] performed an 8-week trial with a total of 300 common carp (initial weight $2.32 \pm 0.02$ g) fed a basal diet supplemented with 2, 3, 4 or 5 g of XMX-1 ($10^8$ CFU/ml)/kg of diet, obtaining as main results the improvement of the intestinal and hepatic health of the common carp, and in other species such as zebrafish, where Xie *et al*., [20] in a 4-week experiment using 240 one-month-old adult zebrafish (initial weight $0.28 \pm 0.003$ g), fed a basal diet supplement with 10 g of XMX-1/kg of diet at a concentration of $10^8$ CFU/ml and after the fourth week they added SVCV to reach $10^6$ TCID50/ml by bath immersion, obtaining important results such as the improvement of the intestinal and hepatic health of the zebrafish and the alteration of the composition of the intestinal microbiota and the protection of the zebrafish from SVCV infection.

Despite these results, the underlying mechanisms remain poorly understood. The exploration of anaerobic bacteria, such as *Cetobacterium*, as potential modulators of host immunity is still in its early stages, and further research is needed to clarify their role and potential as functional additives in aquaculture.

The transcriptomic analysis of tilapia treated with feed supplemented with *Cetobacterium* sp. nov C33 for five days revealed a complex modulation of gene expression, with both upregulated and downregulated pathways contributing to the immune and physiological response. While several immune-related genes were overexpressed, including those involved in apoptosis (cystatin-B), pathogen recognition (beta-galactoside-binding lectin), and inflammation (C-reactive protein), a notable overexpression of key immune system components was observed, particularly in the complement pathway, showing a more complex interaction with immune function that prevents the triggering of harmful inflammation, promoting a balanced immune state.

Likewise, beta-galactoside-binding lectin, which plays a key role in carbohydrate pattern recognition in pathogens, was overexpressed. This lectin family is known for mediating interactions between host immune cells and invading microorganisms, suggesting an enhancement in pathogen recognition capacity, and the activation of immune cells facilitating phagocytosis and inflammation [52]. Furthermore, the transcription factor gene BTF3 Homolog 4 is involved in protein synthesis and regulation, indicating an enhanced transcriptional response, that may contribute to the increased production of immune-related proteins, thereby enhancing immune defense against potential pathogens.

Regarding the apoptotic pathway, the Cystatin B gene was overexpressed, which is a cysteine protease inhibitor involved in the immune response, with a role in viral replication [53]. This recombinant Cystatin B in the case of the red piranha (*Pygocentrus nattereri*) demonstrated potent bacteriostatic activity, indicating its potential use in aquaculture for disease control [54].

Also, increased expression of C-reactive protein (CRP), a marker of acute inflammation and immune activation was found. It plays roles in both innate and adaptive immune responses such as bacterial agglutination, phagocytosis, and

modulation of inflammation during bacterial infection [55,56]. The recombinant CRP of the black rockfish (*Sebastes schlegelii*) exhibited antibacterial activity against *Escherichia coli* and *Streptococcus iniae* [57].

Additionally, genes related to tight junctions such as Claudin-8-like were upregulated, therefore, *Cetobacterium* sp. nov C33 may contribute to intestinal barrier integrity, which is essential for preventing pathogen invasion, a function that could explain the improved gut health and resistance to infection observed in previous feeding trials [58]. Claudin-8-like proteins are required for paracellular chloride transport in the kidney, which modulates the balance of epithelial ion and water transport in the gills, renal tubules, and intestinal epithelial cells, thereby activating the body fluid homeostasis mechanism in fish [59]. Additionally, genes associated with signaling pathways, such as Sodium channel subunit beta-3 and Hemicentin-2, could contribute to overall immune robustness. This suggests enhanced cellular communication and tissue stability. Furthermore, solute carrier family 41 member 1 indicates enhanced signaling and ion transport, which are essential for maintaining cellular homeostasis during immune responses. These metabolic adjustments and the upregulated genes involved in energy production (e.g., NADH dehydrogenase), suggest that *Cetobacterium* sp. nov C33 supports increased metabolic activity to meet the energy demands of an activated immune system.

Additionally, the upregulation of LINGO3, a tumor suppressor and negative regulator of several receptor tyrosine kinases [60], and PYD domains-containing protein 3-like gene, associated with inflammasome function [61], which is normally activated in response to intracellular microbes [62], may suggest that *Cetobacterium* sp. nov C33 could modulate inflammation through this pathway, priming fish for an enhanced immune response.

The upregulation of class I histocompatibility antigen indicates that *Cetobacterium* C33D supplementation can promote the entry of foreign antigens into T cells, enhancing the ability of fish to recognize and eliminate intracellular pathogens [63]. This is crucial in the context of bacterial infections, where cell-mediated immunity plays a dominant role.

Likewise, it was observed that the C33D diet, induced the downregulation of genes related to the immune system, such as the Perforin_1 gene, which is involved in the Granule-exocytosis pathway [64], a process used by cytotoxic lymphocytes to kill abnormal cells. Similarly, a significant reduction in the expression of E3 ubiquitin/ISG15 ligase TRIM25 is associated with various cellular processes, including immune response and cancer, can induce ligand-linked ubiquitination, which activates the signaling pathway to elicit host antiviral innate immunity [65]. Additionally, a reduction in the expression of complement pathway genes, which play a fundamental role in innate immunity, actively participates in the elimination of microorganisms and apoptotic debris as well as the processing of immune complexes [52]. Previous studies indicate that complement is a double-edged sword; although it usually protects the host, it may cause tissue damage when dysregulated or overactivated. The activation is an important component of many diseases [52,60]. Therefore, regulating of the cascade principles is necessary to maintain homeostasis in the host. Conversely, while the immune system of fish is adept at maintaining homeostasis, it remains vulnerable to environmental changes and stressors, which can lead to increased disease susceptibility. A balanced immune response is necessary for good fish health and productivity, preventing the animal from being overwhelmed. As in the case of systemic inflammation, an explosive response could negatively affect the host and growth performance [66]. *Cetobacterium* sp. nov C33 can positively contribute to fish health by balancing immune activation with energy efficiency can be supported by the observed modulation of immune and metabolic pathways together. Our work provides interesting foundations for further studies testing this hypothesis in larger juvenile and adult fishes populations. Additionally, the downregulation of key complement components highlights that this bacterial supplementation does not just "boost" the immune system, but fine-tunes it, potentially preventing overactivation, and could imply a shift in immune strategy, where alternative pathways, such as enhanced cellular immunity or modulation of inflammation through other means (e.g., increased Cystatin-B and PYD domain proteins), become more prominent. The downregulation of these genes may help prevent the overactivation of the immune system, thereby minimizing tissue damage from excessive inflammation, a mechanism often observed in immune homeostasis which is critical for sustainable health management in fish populations.

Likewise, downregulation of metabolic pathways of genes involved in proteolysis, such as alpha-2-macroglobulin-like enzyme, chymotrypsinogen A, carboxypeptidase B1, and trypsin-2, could imply a shift towards controlled protein degradation, which could help prevent excessive tissue damage and maintain protein homeostasis. This could be aligned with the upregulation of immune regulatory genes such as cystatin-B, further suggesting a coordinated response to maintain immune balance and prevent excessive inflammation or proteolysis-induced tissue damage.

In this sense of energy efficiency, downregulation of signaling-related genes, such as transmembrane serine protease 6 and Prestalk-like protein, may reflect a temporary reduction in specific signaling cascades, potentially to conserve energy for other immunological or metabolic processes. Likewise, the downregulation of 18S ribosomal RNA suggests a transient decrease in overall protein synthesis, which may reflect the early stage of immune response, where energy is redirected towards more critical functions, such as immune modulation and cellular defense. Interestingly, genes related to carbohydrate metabolism, including pancreatic alpha-amylase and mammalian acidic chitinase, were also downregulated, which may suggest a reduction in digestive enzyme production, possibly as a compensation to support immune function. It also aligns with the modulation of energy pathways seen in the upregulation of genes such as NADH dehydrogenase, which supports energy production for immune responses rather than digestion. Adding to this, the downregulation of myomesin 3, a gene involved in maintaining myofibrillar structure, could indicate reduced muscle activity during immune activation, as the body reallocates resources towards the immune system. Similarly, downregulation of the 3G regulatory subunit of protein phosphatase 1, which is involved in glycogen synthesis, could reflect a temporary reduction in energy storage, aligning with the increased metabolic demands of an active immune response.

Although our study demonstrated that *Cetobacterium* sp. nov C33 had a positive influence on the balance of the gut microbiota, the current transcriptomic analysis reveals a more complex interaction with the host immune system. Upregulation of immune defense genes and metabolic pathways may mean that *Cetobacterium* sp. nov C33 enhances immune function without triggering harmful inflammation. This finding is in agreement with previous research on the role of microbiota in modulating immune responses in teleost fish, where beneficial bacteria promote a balanced immune state [18].

Though, downregulation of the complement system suggests that caution should be exercised when considering *Cetobacterium* sp. nov C33 as a definitive probiotic, as it may have context-dependent effects. Temporary suppression of the complement pathway could be protective against short-term immune overreaction but further investigation is required to ensure long-term benefits and safety. Overall, the observed immune modulation supports the potential of *Cetobacterium* sp. nov C33 as a functional feed additive in aquaculture, particularly for its ability to enhance immune priming and prevent excessive inflammation. The probiotic effects of *Cetobacterium* sp. nov C33 could be observed in the brief period of supplementation tested. Consequently, additional research is required to verify its function as a probiotic in the long term and higher scale to assess its safety, stability, and extended efficacy in tilapia.

Furthermore, *in vivo* testing in juveniles and adults is necessary to better understand its effects during different rearing periods. The trial was conducted in a controlled laboratory, which may not fully reflect the complexity and environmental variability of commercial aquaculture systems. Indeed, a challenge assay to directly measure protection against specific pathogens or disease resistance under real-world conditions, including metagenomic and metabolomic analyses, could be key to future discoveries in this area and to further understanding of functional changes in microbial activity or interactions with this novel isolate.

## Conclusion

The present study shows that *Cetobacterium* sp. nov C33 can be considered a promising candidate as an anaerobic probiotic for tilapia. Dietary supplementation with *Cetobacterium* sp. nov C33 in Nile tilapia for five days modified the richness and prevalence of the microbiota of the phyla Fusobacteriota and Planctomycetota, and mainly the bacterial genera *Xanthobacter* and *Cetobacterium*. Furthermore, the observed immunomodulatory effects, both in the upregulation and downregulation of key immune and metabolic genes, highlight the potential of the bacteria as part of a comprehensive health

management strategy in aquaculture. While evidence supports its use as a beneficial additive, and there are promising insights about its positive effects in the microbiome composition and the immune response of the host, further research is needed to fully evaluate its probiotic potential and determine the long-term effects of dietary supplementation.

## Supporting information

**S1 Table. Differentially up-regulated genes.** Differentially up-regulated genes from *Oreochromis niloticus* following *Cetobacterium* sp. nov C33 ($p < 0.05$), in kidney samples of tilapia fingerlings from the group fed with the *Cetobacterium* diet (C33D): basal diet with 1 x $10^8$ CFU/g of *Cetobacterium* sp. nov C33, compared to the control group fed with Control Diet (CD): basal diet; after ceasing consumption of *Cetobacterium* Diet for 5 days.
(DOCX)

**S2 Table. Differentially down expressed genes.** Differentially down expressed genes from *Oreochromis niloticus* following *Cetobacterium* sp. nov C33 Diet ($p < 0.05$), in kidney samples of tilapia fingerlings from the group fed with the *Cetobacterium* diet (C33D): basal diet with 1 x $10^8$ CFU/g of *Cetobacterium* sp. nov C33, compared to the control group fed with Control Diet (CD): basal diet; after ceasing consumption of *Cetobacterium* Diet for 5 days.
(DOCX)

**S1 File.** ASV_Ceto.
(CSV)

**S2 File.** Meta_Ceto.
(CSV)

**S3 File.** Taxa_Ceto.
(CSV)

**S4 File.** Data Metataxonomy.
(XLSX)

**S5 File.** Summary_file.
(XLSX)

**S6 File.** Data.
(XLSX)

**S7 File.** Desing.
(TXT)

**S8 File.** Metadata RNAseq.
(XLSX)

## Acknowledgments

The authors are grateful for the great technical support of Jorge Alberto Rodriguez Orjuela and his great experience in handling anaerobic bacteria in vitro. The authors thank the fish farmers for donating the fish specimens.

## Author contributions

**Conceptualization:** Mario Andrés Colorado Gómez, Ruth Yolanda Ruíz Pardo, Luisa Marcela Villamil Díaz.
**Data curation:** Mario Andrés Colorado Gómez, Javier Fernando Melo-Bolívar, Howard Junca, Juan F. Alzate.

**Formal analysis:** Mario Andrés Colorado Gómez, Luisa Marcela Villamil Díaz.

**Funding acquisition:** Mario Andrés Colorado Gómez, Luisa Marcela Villamil Díaz.

**Investigation:** Mario Andrés Colorado Gómez, Luisa Marcela Villamil Díaz.

**Methodology:** Mario Andrés Colorado Gómez, Luisa Marcela Villamil Díaz.

**Project administration:** Mario Andrés Colorado Gómez, Luisa Marcela Villamil Díaz.

**Resources:** Mario Andrés Colorado Gómez, Luisa Marcela Villamil Díaz.

**Software:** Mario Andrés Colorado Gómez, Javier Fernando Melo-Bolívar, Howard Junca, Juan F. Alzate, Luisa Marcela Villamil Díaz.

**Supervision:** Ruth Yolanda Ruíz Pardo, Luisa Marcela Villamil Díaz.

**Validation:** Mario Andrés Colorado Gómez, Ruth Yolanda Ruíz Pardo, Luisa Marcela Villamil Díaz.

**Visualization:** Mario Andrés Colorado Gómez.

**Writing – original draft:** Mario Andrés Colorado Gómez.

**Writing – review & editing:** Mario Andrés Colorado Gómez, Luisa Marcela Villamil Díaz.

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
