## [Decision Letter · Decision Letter 0]

13 Jun 2025

PONE-D-25-18938Anaerobic bacteria Cetobacterium sp. nov C33 plays a crucial role in the intestinal microbial balance and regulation of gene expression to immune and metabolic responses in Nile TilapiaPLOS ONE

Dear Dr. Villamil Díaz,

Thank you for submitting your manuscript to PLOS ONE. After careful consideration, we feel that it has merit but does not fully meet PLOS ONE’s publication criteria as it currently stands. Therefore, we invite you to submit a revised version of the manuscript that addresses the points raised during the review process.

If applicable, we recommend that you deposit your laboratory protocols in protocols.io to enhance the reproducibility of your results. Protocols.io assigns your protocol its own identifier (DOI) so that it can be cited independently in the future. For instructions see: https://journals.plos.org/plosone/s/submission-guidelines#loc-laboratory-protocols. Additionally, PLOS ONE offers an option for publishing peer-reviewed Lab Protocol articles, which describe protocols hosted on protocols.io. Read more information on sharing protocols at . Additionally, PLOS ONE offers an option for publishing peer-reviewed Lab Protocol articles, which describe protocols hosted on protocols.io. Read more information on sharing protocols at https://plos.org/protocols?utm_medium=editorial-email&utm_source=authorletters&utm_campaign=protocols..

We look forward to receiving your revised manuscript.

Kind regards,

Amel Mohamed El Asely

Academic Editor

PLOS ONE

Journal Requirements:

2. To comply with PLOS ONE submissions requirements, in your Methods section, please provide additional information regarding the experiments involving animals and ensure you have included details on (1) methods of sacrifice, and (2) efforts to alleviate suffering.

3. Thank you for stating the following financial disclosure: [This research was funded by the Sistema Nacional de Regalias (SGR): Ministerio de Hacienda de Colombia y Ministerio de Ciencias y Tecnología e Innovación de Colombia. Project code before the National Bank of Investment Programs and Projects bpin: 2020000100487. The project was carried out by the Universidad de La Sabana and the Servicio Nacional de Aprendizaje -SENA- Regional La Guajira.].

4. In the online submission form, you indicated that [Data will be made available on request.].

6. We notice that your supplementary tables are included in the manuscript file. Please remove them and upload them with the file type 'Supporting Information'. Please ensure that each Supporting Information file has a legend listed in the manuscript after the references list.

Reviewers' comments:

Reviewer's Responses to Questions

**Comments to the Author**

1. Is the manuscript technically sound, and do the data support the conclusions?

Reviewer #1: Yes

Reviewer #2: Partly

Reviewer #3: Yes

Reviewer #4: Yes

2. Has the statistical analysis been performed appropriately and rigorously? 

Reviewer #1: No

Reviewer #2: N/A

Reviewer #3: Yes

Reviewer #4: Yes

3. Have the authors made all data underlying the findings in their manuscript fully available?

Reviewer #1: Yes

Reviewer #2: No

Reviewer #3: Yes

Reviewer #4: Yes

4. Is the manuscript presented in an intelligible fashion and written in standard English?

Reviewer #1: Yes

Reviewer #2: Yes

Reviewer #3: No

Reviewer #4: Yes

5. Review Comments to the Author

Reviewer #1: Major comments:

Even if the current study offers an effective probiotic to improve fish development, immunity, and reproduction, it has numerous limitations that should be taken into account, including the following implications

1- The probiotic effects of Cetobacterium sp. nov C33 were evaluated over a short supplementation period, which limits understanding of its long-term safety, stability, and sustained efficacy in tilapia.

2- The study was conducted exclusively on fingerlings, so the applicability of findings to juveniles or adults remains uncertain.

3- The trial was conducted under controlled laboratory or hatchery settings, which may not fully capture the complexity and environmental variability of commercial aquaculture systems.

4- Although immune-related gene expression was assessed, the study did not include a disease challenge to directly measure protection against specific pathogens or real-world disease resistance.

5- While 16S rRNA sequencing provided taxonomic shifts in microbiota composition, it did not include metagenomic or metabolomic analyses to clarify functional changes in microbial activity or interactions.

6- Findings are specific to Nile tilapia, and their generalizability to other aquaculture species is unknown without further comparative studies.

So, it is highly recommended to have a paragraph to discuss the major limitations of the present study before the section of the Conclusion.

Minor comments:

1- English proofreading is highly recommended.

2- No statistical analysis, so please analyse that data.

3- No line numbers.

4- Please prepare the manuscript according to Plos One's instructions. For example the title page does not follow the guidelines.

5- Check the grammer the 2nd paragraph in the introduction “The intestinal bacterial composition in tilapia is has been previously described”.

6- Please add brackets for SVCV to indicate that this is abbrreviatoon for spring viraemia of carp virus (3rd paragraph in the introduction)

7- Check the grammer the 2nd paragraph in the introduction “y Though the search or new potential probiotics” I believe you mean “through the search for new potential probiotics”, so please correct it.

8- The references you used to plan your experiments are missing from the entire methods section; please update and add them across the entire manuscript.

9- Keep the conclusion to a minimum to make it more representative and focused

Reviewer #2: Peer Reviewer s Comments

The Paper entitled Anaerobic bacteria Cetobacterium sp. nov C33 plays a crucial role in the intestinal microbial balance and regulation of gene expression to immune and metabolic responses in Nile Tilapia which has been submitted to publish in PLOS One presents a good investigative efforts but the hypothesis derived from the obtained data is scientifically and logically wrong very likely because there is a gap in present understanding on knowledge.

Low cell population e.g.108cells/g of Anaerobic bacteria Cetobacterium sp. nov C33 present in feed are practically too less to enable the fed innoculum to appear as dominating species in Nile Tilapia particularly in the case when it requires anaerobic condition to grow at 28C whereas the temperature of fish body is usually lower than 28C with limited availability of oxygen. However these cells by virtue of their high hydrophobicity can make clusters having potential to adhere with cells membrane of cells lining the gut effecting Oxygen, cholesterol and different metallic ions transport particularly Copper and Iron across the cell membranes effecting the host cells functions of target cells and also interconnected cells translating the impact across the organs connected through body systems modulating genes network operations by virtue of copper and iron balance in associated and dissociated forms effected differently by oxygen availability at cellular level, enabling cells to attain new phenotypes with altered homeostasis supported environment leading to the shift in microbial populations and shift in body cell functions over the period attaining altered normal. In response of stress conditions in gut the anaerobic bacteria Cetobacterium sp. nov C33 produce pigments and other secondary metabolites e.g. chelating agents enzymes etc which alters the iron availability and its accessibility in the cell environment which is is evident by up regulation and down regulation of different genes without activating an inflammatory pathway inspite of involving immune system of the host and without uncontrolled fluctuation in energy demand in body cells subsets.

Using yeast as a model, it has already been established that CTR1,(CTR1 gene encodes high affinity copper transporter) SOD1, superoxide dismutase 1 gene, GSH encoding genes and cholesterol cellular biosynthesis genes which are interdependently co-regulated at transcriptional level to sustain metallic ions homeostasis, energy homeostasis, cell s aging leading to cell evolution or cell death. Papers revealing the underlying molecular mechanisms have already been submitted.

The work presented in the paper seems to be very exciting but the hypothesis drawn is wrong very likely because of the gap in the knowledge.

I would recommend publishing this piece of work in PLOS one after revising the hypothesis presented in the paper on the ground mentioned above after making through independent critical analysis of published data and my published and unpublished but submitted work which can be obtained from Professor Simon Avery University of Nottingham United Kingdom by contacting him on e-mail Simon.Avery@nottingham.ac.uk if he kindly agrees to do so.

Helpful References

1) Bishop, A.L, Rab, F.A, Sumner, E.R, Avery, S.V (2007). Phenotypic heterogeneity can enhance rare-cell survival in 'stress-sensitive' yeast populations. Mol Microbiol. 63(2):507-20. doi: 10.1111/j.1365-2958.2006.05504.x. Epub 2006 Dec 14. PMID: 17176259

2) Rab, F.A. Solving a biological problem: Research methodologyVDM Verlag Dr. Müller (April 7, 2010) ISBN-10 ‏ : ‎ 9783639247497 ISBN-13 ‏ : ‎ 978-3639247497

3) Rab, F.A.(2014)Environmentally Modulated Evolution through Genetic Regulation Information Systems for Biotechnology ISB News Reports June/July

4) Rab, F.A.(2015) A Global Prospective on Research. Academe September October

5) Rab, F.A.(2016) Biotechnology and its Potentials and Challenges

EC Agriculture 3.4 : 705-707

6)Rab, F.A.(2017)(a)Who Should Come in Research? EC Nutrition 6.3:102-104

7) Rab, F.A. (2017)(b) Eat Fresh Live Young EC Nutrition RCO.01:03-05

8) Rab, F.A(2018)(a) Is Sugar an Accessory or a Neccessary.EC Nutrition 13.4:236-237

9) Rab, F.A.(2018)(b)Drug Disease Relationship and the Role of Food in Healthy Living.EC Nutrition 13.8 doing the bold one

10) Rab, F.A.(2018)(c)Genome-Nutrifortified Diets-Their Disease Protection and Remedy Potential. J.Prob Health 6 :204 doi: 10.4172/2329-8901.1000204

Rab FA (2018) Genome-Nutrifortified Diets-Their Disease Protection and Remedy Potential. J Prob Health 6: 204. doi:

10.4172/2329-8901.1000

Rab FA (2018) Genome-Nutrifortified Diets-Their Disease Protection and Remedy Potential. J Prob Health 6: 204. doi:

10.4172/2329-8901.1000

11) Rab, F.A.(2019)(a) Comparison between Safety Risks Associated with Domestically Processed Food and Commercially Manufactured Processed Food across the Food Supply Chain.EC Nutrition 14.5:414-416.

12) Rab, F.A(2019)(b) Is Hunger More Dangerous than having Mal-Nutrition or Consuming Unsafe Diet.EC Nutrition 14.12:01-05

13) Rab, F.A.(2020)(a) Hurdles in Progression of Knowledge and its Global Impact. Glob J Res Rev Vol.7 No.1:47. DOI: 10.36648/2393-8854.7.1.47

14) Rab, F.A. (2020)(b) Halal or Haram-New for Religious Scholars Muslim World and Food Supply Chain Stake Holders. Int J Nutr Sci & Food Tech. 6:3

15) Rab, F.A. (2021) Food items biologically tailored to meet nutritional deficiency challenge during Covid 19 Pandemic. J Prob Health. 9:233.

16) Abdur Rab, F., Hassan, A. (2022). Tourism, Health Promoting Food Domain and Technology Applications: Individual’s Genes Reservoir, Environmental Change and Food in Natural Health Context. In: Hassan, A. (eds) Handbook of Technology Application in Tourism in Asia. Springer, Singapore. https://doi.org/10.1007/978-981-16-2210-6_53

17) Rab et al.,(Co authored with Dr Sami Farooq, Dean of School of Management Sciences, Ghulam Ishaq Khan Institute of Engineering Sciences and Technology Swabi Pakistan) (2023) COVID 19- Technology driven Economically coordinated Supply Chain Chaos, its Remedy and Recommendations [Complete Paper and Science Policy Brief had been discussed in World Health Organization (WHO) s Meeting held on 4th May 2023 and in United Nation (UN) s Science Forum Meeting ( Science Technology and Innovation Forum e.g. STI Forum) held on 3-4 May 2023 in United States of America (USA) on the author (Dr Faiza Abdur Rab) s request which she made in capacity of Reviewer for all the Research Journals being published by The Royal Society of Chemistry. These papers are available on request].

18)Rab, F.A.(2023) Would Science Knowledge, Food and Agriculture Sector be Considered Basic Human Services or Commercial Services in this Post Covid19 Era? Decision will Determine the Destination. EC Nutrition 18.10: 01-08

19) Rab, F.A. (2023) Corona Virus (SARS-CoV-2 ) Rewinding Models in Science and Business and Redefining Illness Manifestations [submitted in Kuwait Journal of Science by Elsevier]

(20) Rab F.A (2023) Covid 19 Virus- Genomic and Molecular Gaming and the Role of Diet Preparations in Therapy [ under review in one of the well reputed Journals by Springer Nature]

21) Rab, F.A. (2024) Universal Evolutionary capacitor switch operation regulation gimmicks driving illness and natural cure [ under process in one of the well reputed Journals by Nature]

22) Colorado Gómez, M. A., Melo-Bolívar, J. F., Ruíz Pardo, R. Y., Rodriguez, J. A., & Villamil, L. M. (2023). Unveiling the Probiotic Potential of the Anaerobic Bacterium Cetobacterium sp. nov. C33 for Enhancing Nile Tilapia (Oreochromis niloticus) Cultures. Microorganisms, 11(12), 2922. https://doi.org/10.3390/microorganisms11122922

23) Chatfield, C. H., & Cianciotto, N. P. (2007). The secreted pyomelanin pigment of Legionella pneumophila confers ferric reductase activity. Infection and immunity, 75(8), 4062–4070. https://doi.org/10.1128/IAI.00489-07

24) Maret W.(2024) The Extracellular Metallometabolome: Metallophores, Metal Ionophores, and Other Chelating Agents as Natural Products. Natural Product Communications.;19(8). doi:10.1177/1934578X241271701

Reviewer #3: This manuscript presents a timely and relevant study addressing a critical need in the aquaculture industry. As one of the fastest-growing sectors in global food production, aquaculture—particularly the farming of Nile tilapia (Oreochromis niloticus)—faces increasing challenges due to rising consumer demand and the resulting intensification of production systems. These pressures often lead to compromised fish health and increased disease prevalence.

However, I have significant concerns regarding the overall presentation and writing style of the manuscript. It does not adhere to the formatting and structural guidelines typically expected for scientific manuscripts, such as those outlined by PLOS ONE. The writing style resembles that of a paper prepared for a different discipline, rather than aligning with standard scientific conventions. Additionally, the English language requires substantial editing for clarity, grammar, and readability to meet the standards of a peer-reviewed journal.

In the Materials and Methods section, please provide detailed information on how Cetobacterium sp. nov C33 was cultured under anaerobic conditions, including the specific broth medium used and any relevant growth parameters.

To improve transparency and enhance the manuscript’s clarity for readers in the field, I recommend rewriting sentences like: 'The gut-associated microbiota modulation was studied using culture independent techniques by 16S amplicons Illumina sequencing.' As currently phrased, it is difficult to understand. Consider rewording in a clear and concise manner, such as: 'The composition of gut-associated microbiota was analyzed using culture-independent 16S rRNA gene sequencing via Illumina technology.' This helps ensure that the methods and findings are easily understood and accessible to a broader scientific audience.

To improve transparency and ensure a clearer understanding, please clarify how microbial diversity was calculated and how the data were normalized. The statement, 'To estimate microbial diversity, abundance tables were normalized with Total Sum Scaling (TSS) in each sample,' lacks sufficient detail. It would be helpful to specify the exact steps used in the normalization process, including how TSS was applied and whether diversity metrics were calculated on raw or normalized data. Providing this information will strengthen reproducibility and comprehension for readers.

Reviewer #4: The abstract could be a little larger and elaborate. The overall writing is average not up to the mark.It needs more grammartical refining. More references should be used to validate the given informations properly.

6. PLOS authors have the option to publish the peer review history of their article (what does this mean?). If published, this will include your full peer review and any attached files.). If published, this will include your full peer review and any attached files.

.

Reviewer #1: No

Reviewer #2: **Yes:**Faiza Abdur RabFaiza Abdur Rab

Reviewer #3: No

Reviewer #4: **Yes:**Md. Afif UllahMd. Afif Ullah

While revising your submission, please upload your figure files to the Preflight Analysis and Conversion Engine (PACE) digital diagnostic tool, https://pacev2.apexcovantage.com/. PACE helps ensure that figures meet PLOS requirements. To use PACE, you must first register as a user. Registration is free. Then, login and navigate to the UPLOAD tab, where you will find detailed instructions on how to use the tool. If you encounter any issues or have any questions when using PACE, please email PLOS at . PACE helps ensure that figures meet PLOS requirements. To use PACE, you must first register as a user. Registration is free. Then, login and navigate to the UPLOAD tab, where you will find detailed instructions on how to use the tool. If you encounter any issues or have any questions when using PACE, please email PLOS at figures@plos.org. Please note that Supporting Information files do not need this step.. Please note that Supporting Information files do not need this step.

---

## [Author Response · Author response to Decision Letter 1]

12 Aug 2025

Response to Reviewer

Thank you very much for taking the time to review this manuscript. Please find the detailed responses below and the corresponding revisions and changes in the re-submitted files.

Response to Reviewer 1 Comments

Comments 1: Even if the current study offers an effective probiotic to improve fish development, immunity, and reproduction, it has numerous limitations that should be taken into account, including the following implications

1- The probiotic effects of Cetobacterium sp. nov C33 were evaluated over a short supplementation period, which limits understanding of its long-term safety, stability, and sustained efficacy in tilapia.

Response: Short-term administration of microorganisms can also offer advantages in terms of investment in treatment, especially when positive results are found. However, the study of long-term administration will also be very interesting to analyze in the longer term.

2- The study was conducted exclusively on fingerlings, so the applicability of findings to juveniles or adults remains uncertain.

Response: The research focuses on fingerlings due to their susceptibility to disease. Furthermore, differences in the microbiota at different culture stages have not been studied, so evaluating the effect of Cetobacterium sp. nov C33 at other culture stages and on a larger scale is very interesting and could be done in future studies.

3- The trial was conducted under controlled laboratory or hatchery settings, which may not fully capture the complexity and environmental variability of commercial aquaculture systems.

Response: Agreed, most probiotic studies are conducted on a laboratory scale rather than in production settings. This is related to ensuring the safety of the microorganisms being evaluated, especially Cetobacterium sp. nov C33, which has been little studied. Furthermore, scaling up production of Cetobacterium sp. nov C33 for commercial-scale testing requires much more research, but given the positive results, we want to move forward in this direction.

4- Although immune-related gene expression was assessed, the study did not include a disease challenge to directly measure protection against specific pathogens or real-world disease resistance.

Response: At the time of the research, there was no permission from the institutional ethics committee to conduct the experimental challenge.

5- While 16S rRNA sequencing provided taxonomic shifts in microbiota composition, it did not include metagenomic or metabolomic analyses to clarify functional changes in microbial activity or interactions.

Response:

6- Findings are specific to Nile tilapia, and their generalizability to other aquaculture species is unknown without further comparative studies.

Response: Our findings are specific to Nile tilapia and we do not generalize them to other species.

7- So, it is highly recommended to have a paragraph to discuss the major limitations of the present study before the section of the Conclusion.

Response: We include the study's limitations in a paragraph in Line 749-762. We appreciate the peer reviewers' observations, as they highlight new opportunities to continue our research.

Minor comments:

1- English proofreading is highly recommended.

Response: The authors have revised the manuscript according to the reviewers’ comments, and the quality is improved. However, if the editor deems it appropriate, after the review we would hire a writing revision service as recommended by the journal.

2- No statistical analysis, so please analyze that data.

Response: Statistical analysis was carried out in included in the submitted manuscript, nevertheless the methodology was improved to described the analysys carried out. Also, we included in the Tables 1 and 2 the Statistical data for differential significantly for genera taxon, Furthermore, in supplementary tables S1 and S2, we included the Statical data for differential significantly for genes upregulated and downregulated.

3- No line numbers.

Response: We applied the Line numbers according to Plos One's instructions.

4- Please prepare the manuscript according to Plos One's instructions. For example the title page does not follow the guidelines.

Response: We applicated the guidelines of Plos One´s instructions

5- Check the grammer the 2nd paragraph in the introduction “The intestinal bacterial composition in tilapia is has been previously described”.

Response: We correct grammar and wrote in Line 57-59: “Previous research has indicated that the intestinal bacterial composition of tilapia is primarily composed of three phyla: Fusobacteria, Bacteroidota, and Proteobacteria”.

6- Please add brackets for SVCV to indicate that this is abbreviation for spring viraemia of carp virus (3rd paragraph in the introduction)

Response: We added brackets for SVCV in Line 70.

7- Check the grammer the 2nd paragraph in the introduction “y Though the search or new potential probiotics” I believe you mean “through the search for new potential probiotics”, so please correct it.

Response: We wrote “through the search for new potential probiotics” in the Line 97-98.

8- The references you used to plan your experiments are missing from the entire methods section; please update and add them across the entire manuscript.

Response: We included the references in the methods section. The references are:

9. Melo-Bolívar, J. F., Ruiz Pardo, R. Y., Quintanilla-Carvajal, M. X., Díaz, L. E., Alzate, J. F., Junca, H., Rodríguez Orjuela, J. A., & Villamil Diaz, L. M. (2023). Evaluation of dietary single probiotic isolates and probiotic multistrain consortia in growth performance, gut histology, gut microbiota, immune regulation, and infection resistance of Nile tilapia, Oreochromis niloticus, shows superior monostrain performance. Fish & Shellfish Immunology, 140, 108928. https://doi.org/10.1016/J.FSI.2023.108928

19. FA, R. (2018). Genome-Nutrifortified Diets-Their Disease Protection and Remedy Potential. Journal of Probiotics & Health, 06(02). https://doi.org/10.4172/2329-8901.1000204

22. Kilkenny, C., Browne, W. J., Cuthill, I. C., Emerson, M., & Altman, D. G. Improving bioscience research reporting: the ARRIVE guidelines for reporting animal research. PLoS biology. (2010; 8(6), e1000412. https://doi.org/10.1371/journal.pbio.1000412

23. Rose, M., Everitt, J., Hedrich, H., Schofield, J., Dennis, M., Scott, E., & Griffin, G. (2013). ICLAS working group on harmonization: International guidance concerning the production care and use of genetically-altered animals. Laboratory Animals, 47(3), 146–152. https://doi.org/10.1177/0023677213479338

25. Sharifuzzaman, S. M., & Austin, B. (2010). Development of protection in rainbow trout (Oncorhynchus mykiss, Walbaum) to Vibrio anguillarum following use of the probiotic Kocuria SM1. Fish & Shellfish Immunology, 29(2), 212–216. https://doi.org/10.1016/J.FSI.2010.03.008

31. Zhou, R., Ng, S. K., Sung, J. J. Y., Goh, W. W. bin, & Wong, S. H. (2023). Data pre-processing for analyzing microbiome data – A mini review. Computational and Structural Biotechnology Journal, 21, 4804–4815. https://doi.org/10.1016/J.CSBJ.2023.10.001

32. Cao, Q. X., Sun, X., Rajesh, K., Chalasani, N., Gelow, K., Katz, B., Shah, V. H., Sanyal, A. J., & Smirnova, E. (2020). Effects of microbiome rare taxa filtering on statistical analysis. https://doi.org/10.21203/RS.3.RS-34781/V1

33. Cameron, E. S., Schmidt, P. J., Tremblay, B. J. M., Emelko, M. B., & Müller, K. M. (2021). Enhancing diversity analysis by repeatedly rarefying next generation sequencing data describing microbial communities. Scientific Reports, 11(1). https://doi.org/10.1038/S41598-021-01636-1

34. Cappellato, M., Baruzzo, G., & Camillo, B. di. (2022). Investigating differential abundance methods in microbiome data: A benchmark study. PLoS Computational Biology, 18(9), e1010467. https://doi.org/10.1371/JOURNAL.PCBI.1010467

9- Keep the conclusion to a minimum to make it more representative and focused

Response: We reduce the conclusion chapter. Lines: 764-774.

Response to Reviewer 2 Comments

Comment 1: The Paper entitled Anaerobic bacteria Cetobacterium sp. nov C33 plays a crucial role in the intestinal microbial balance and regulation of gene expression to immune and metabolic responses in Nile Tilapia which has been submitted to publish in PLOS One presents a good investigative efforts but the hypothesis derived from the obtained data is scientifically and logically wrong very likely because there is a gap in present understanding on knowledge.

Response 1: We, according to the scope of our project, did not present a hypothesis, we reference the possibility that Cetobacterium could be a candidate to be considered as a probiotic for use in fish. We removed the word hypothesis from the document to avoid confusion for the reader. However, according to the response obtained at the transcriptomic level, we express the possibility that a greater abundance of Cetobacterium isolated from specimens of the same species generates a positive effect on the health of fish.

Comment 2: Low cell population e.g.108cells/g of Anaerobic bacteria Cetobacterium sp. nov C33 present in feed are practically too less to enable the fed innoculum to appear as dominating species in Nile Tilapia particularly in the case when it requires anaerobic condition to grow at 28°C whereas the temperature of fish body is usually lower than 28°C with limited availability of oxygen.

Response 2: Previous studies conducted within our research group, utilizing culture-independent techniques, revealed that Cetobacterium was a predominant genus within the intestinal microbiota, Melo-Bolivar et al. (2019) reported that “The most abundant bacterium in the first days of the CFCEC, was Cetobacterium sp.”. On the other hand, for use the probiotics in Nile tilapia, various authors used bacteria in different quantities, such as Wang et al. (2008) who use Enterococcus faecium 1 × 10 7 UFC/mL during 40 day and achieve final weight increase. Hassaan et al. (2018) used Bacillus subtilis 1,1 × 10 5 UFC/g for 84 day and reach improvement of growth performance. Van Doan et al. (2019) used Lactobacillus plantarum 1 × 10 8 UFC/g during 84 days and reach improvement of growth performance. Finally, Sewaka et al. (2019) used Lactobacillus rhamnosus 1 × 10 8 UFC/g during 2 weeks, and enhancement of histo-morphological intestinal parameters.

References

Hassaan, M.S.; Soltan, M.A.; Jarmołowicz, S.; Abdo, H.S. Combined effects of dietary malic acid and Bacillus subtilis on growth, gut microbiota and blood parameters of Nile tilapia (Oreochromis niloticus). Aquac. Nutr (2018) 24, 83–93

Melo-Bolívar, J. F., Ruiz Pardo, R. Y., Hume, M. E., Nisbet, D. J., Rodríguez-Villamizar, F., Alzate, J. F., Junca, H., & Villamil Díaz, L. M. Establishment and characterization of a competitive exclusion bacterial culture derived from Nile tilapia (Oreochromis niloticus) gut microbiomes showing antibacterial activity against pathogenic Streptococcus agalactiae. PLOS ONE. (2019). 14(5), e0215375.

Sewaka, M.; Trullas, C.; Chotiko, A.; Rodkhum, C.; Chansue, N.; Boonanuntanasam, S.; Pirarat, N. Efficacy of synbiotic Jerusalem artichoke and Lactobacillus rhamnosus GG-supplemented diets on growth performance, serum biochemical parameters, intestinal morphology, immune parameters and protection against Aeromonas veronii in juvenile red tilapia (Oreochromis spp.). Fish Shellfish Immunol (2019). 86, 260–268

Van Doan, H.; Kurian, A.; Hoseinifar, S.H.; Sel-audom, M.; Jaturasitha, S.; Tongsiri, S.; Ringø, E. Dietary inclusion of orange peels derived pectin and Lactobacillus plantarum for Nile tilapia (Oreochromis niloticus) cultured under indoor biofloc systems. Aquaculture (2019). 508, 98–105.

Wang, Y.-B.; Tian, Z.-Q.; Yao, J.-T.; Li, W. Effect of probiotics, Enteroccus faecium, on tilapia (Oreochromis niloticus) growth performance and immune response. Aquaculture (2008). 277, 203–207

Comment 3: However these cells by virtue of their high hydrophobicity can make clusters having potential to adhere with cells membrane of cells lining the gut effecting Oxygen, cholesterol and different metallic ions transport particularly Copper and Iron across the cell membranes effecting the host cells functions of target cells and also interconnected cells translating the impact across the organs connected through body systems modulating genes network operations by virtue of copper and iron balance in associated and dissociated forms effected differently by oxygen availability at cellular level, enabling cells to attain new phenotypes with altered homeostasis supported environment leading to the shift in microbial populations and shift in body cell functions over the period attaining altered normal. In response of stress conditions in gut the anaerobic bacteria Cetobacterium sp. nov C33 produce pigments and other secondary metabolites e.g. chelating agents enzymes etc which alters the iron availability and its accessibility in the cell environment which is is evident by up regulation and down regulation of different genes without activating an inflammatory pathway inspite of involving immune system of the host and without uncontrolled fluctuation in energy demand in body cells subsets. Using yeast as a model, it has already been established that CTR1,(CTR1 gene encodes high affinity copper transporter) SOD1, superoxide dismutase 1 gene, GSH encoding genes and cholesterol cellular biosynthesis genes which are interdependently co-regulated at transcriptional level to sustain metallic ions homeostasis, energy homeostasis, cell s aging leading to cell evolution or cell death. Papers revealing the underlying molecular mechanisms have already been submitted.

Response 3: Cetobacterium sp. nov C33 does not produce pigments and no regulation with metal ions was found, neither in the literature nor through genome review. We applied antiSMASH, a bioinformatics tool that allows the detection of gene clusters associated with the biosynthesis of secondary metabolites, including siderophores. It is used to analyze complete or assembled bacterial genomes and search for genes encoding enzymes involved in siderophore synthesis. Cetobacterium sp. nov C33, we did not find siderophores, but the results showed four regions in the genomes like glycerol dehydrogenase. (Region 1 – NRPS, RFV38_04915 glycerol dehydrogenase Locus tag: RFV38_04915 Protein ID: MDX8335839.1 Gene: None Location: 17,505 - 18,581, (total: 1077 nt)); Region 1 - NRPS-like, Location: 67,328 - 69,838, (total: 2511 nt); Region 1 - RiPP-like Location: 32,143 - 33,093, (total: 951 nt); Region 1 - terpene-precursor. Location: 14,714 - 15,595, (total: 882 nt).

NRPS-like (Nonribosomal Peptide Synthetase-like) and siderophores are both involved in the biosynthesis of molecules with distinct functions, but they differ in their structure, biosynthesis pathways, and specific roles. NRPS-like systems produce a diverse array of bioactive peptides, including antibiotics and immunosuppressants, while siderophores are primarily involved in iron acquisition by binding and transporting ferric iron (Fe3+).

It is important to highlight that Cetobacterium sp nov C33 was isolated from Oreochromis niloticus (Colorado Gómez et al., 2023) and that Cetobacterium has been reported as the most abundant genus in the gut microbiota of Nile tilapia (Melo-Bolívar et al., 2019). Cetobacterium itself has not been reported as an infectious agent in fish (Xiao et al., 2021). I

References

Balcázar JL, Vendrell D, de Blas I, Ruiz-Zarzuela I, Gironé s O, Muzquiz JL. Immune modulation by probiotic strains: Quantification of phagocytosis of Aeromonas salmonicida by leukocytes isolated from gut of rainbow trout (Oncorhynchus mykiss) using a radiolabelling assay. Comp Immunol Microbiol Infect Dis (2006) 29(5):335–43. doi: 10.1016/j.cimid.2006.09.004

Cerezuela R, Guardiola FA, Cuesta A, Esteban MA. Enrichment of gilthead seabream (Sparus aurata L.) diet with palm fruit extract andn probiotics: effects on skin mucosal immunity. Fish Shellfish Immunol (2016) 49:100–9. doi: 10.1016/j.fsi.2015.12.028

Colorado Gómez, M. A., Melo-Bolívar, J. F., Ruíz Pardo, R. Y., Rodriguez, J. A., & Villamil, L. M. Unveiling the Probiotic Potential of the Anaerobic Bacterium Cetobacterium sp. nov. C33

---

## [Decision Letter · Decision Letter 1]

21 Sep 2025

PONE-D-25-18938R1Anaerobic bacteria Cetobacterium sp. nov C33 plays a crucial role in the intestinal microbial balance and regulation of gene expression to immune and metabolic responses in Nile TilapiaPLOS ONE

Dear Dr. Villamil Díaz,

Thank you for submitting your manuscript to PLOS ONE. After careful consideration, we feel that it has merit but does not fully meet PLOS ONE’s publication criteria as it currently stands. Therefore, we invite you to submit a revised version of the manuscript that addresses the points raised during the review process.

If applicable, we recommend that you deposit your laboratory protocols in protocols.io to enhance the reproducibility of your results. Protocols.io assigns your protocol its own identifier (DOI) so that it can be cited independently in the future. For instructions see: https://journals.plos.org/plosone/s/submission-guidelines#loc-laboratory-protocols. Additionally, PLOS ONE offers an option for publishing peer-reviewed Lab Protocol articles, which describe protocols hosted on protocols.io. Read more information on sharing protocols at . Additionally, PLOS ONE offers an option for publishing peer-reviewed Lab Protocol articles, which describe protocols hosted on protocols.io. Read more information on sharing protocols at https://plos.org/protocols?utm_medium=editorial-email&utm_source=authorletters&utm_campaign=protocols..

We look forward to receiving your revised manuscript.

Kind regards,

Amel Mohamed El Asely

Academic Editor

PLOS ONE

Journal Requirements:

Reviewers' comments:

Reviewer's Responses to Questions

**Comments to the Author**

1. If the authors have adequately addressed your comments raised in a previous round of review and you feel that this manuscript is now acceptable for publication, you may indicate that here to bypass the “Comments to the Author” section, enter your conflict of interest statement in the “Confidential to Editor” section, and submit your "Accept" recommendation.

Reviewer #2: (No Response)

2. Is the manuscript technically sound, and do the data support the conclusions?

Reviewer #2: Partly

3. Has the statistical analysis been performed appropriately and rigorously? 

Reviewer #2: No

4. Have the authors made all data underlying the findings in their manuscript fully available?

Reviewer #2: No

5. Is the manuscript presented in an intelligible fashion and written in standard English?

Reviewer #2: Yes

6. Review Comments to the Author

Reviewer #2: Reviewer s comments

The comments of the authors in response of the points I raised have failed to justify the given scientific status of the revised paper.

For instance technically speaking in a common person s language hypothesis is referred as a possibility. Merely substituting the synonyms cannot serve the purpose to make a scientific content authentic

The authors failed to present a scientifically authentic counter argument against my stance " Low cell population e.g.108 cells/g of Anaerobic bacteria Cetobacterium sp. nov C33 present in feed are practically too less to enable the fed inoculums to appear as dominating species in Nile Tilapia particularly in the case when it requires anaerobic condition to grow at 28°C whereas the temperature of fish body is usually lower than 28°C with limited availability of oxygen ".

Furthermore it seems that paper under revision has been written with inadequate literature search reflected as deficient understanding of relevant domains of knowledge. There is no single living entity which does operate without interplay of mineral ions. As far as production of pigments are concerned, genes encoding them may not be present on genome of the bacteria such as they can be present on plasmid or production of pigments can be an induced manifestation of a biological event or change in the environment.

Supporting scientific abstract statement

"microorganisms might evolve responses to sublethal concentrations of these metabolites, either to protect themselves from inhibition or to change certain behaviors in response to the local abundance of another species. Here, we report that violacein production by C. violaceum ATCC 31532 is induced in response to hygromycin A from Streptomyces sp. 2AW, and we show that this response is dependent on inhibition of translational polypeptide elongation and a previously uncharacterized two-component regulatory system. The breadth of the transcriptional response beyond violacein induction suggests a surprisingly complex metabolite-mediated microbe-microbe interaction and supports the hypothesis that antibiotics evolved as signal molecules".

Reference

Lozano, G. L., Guan, C., Cao, Y., Borlee, B. R., Broderick, N. A., Stabb, E. V., & Handelsman, J. (2020). A Chemical Counterpunch: Chromobacterium violaceum ATCC 31532 Produces Violacein in Response to Translation-Inhibiting Antibiotics. mBio, 11(3), e00948-20. https://doi.org/10.1128/mBio.00948-20.

The concerns raised on assessing the submission entitled Anaerobic bacteria Cetobacterium sp. nov C33 plays a crucial role in the intestinal microbial balance and regulation of gene expression to immune and metabolic responses in Nile Tilapia have not been addressed while revising the paper, scientific flaws still sustain in the revision, that builds a ground to recommend major revision of the revised version of the submission while addressing the points raised by me in capacity of reviewer prior publishing the paper.

7. PLOS authors have the option to publish the peer review history of their article (what does this mean?). If published, this will include your full peer review and any attached files.). If published, this will include your full peer review and any attached files.

.

Reviewer #2: **Yes:**Dr Faiza Abdur RabDr Faiza Abdur Rab

While revising your submission, please upload your figure files to the Preflight Analysis and Conversion Engine (PACE) digital diagnostic tool, https://pacev2.apexcovantage.com/. PACE helps ensure that figures meet PLOS requirements. To use PACE, you must first register as a user. Registration is free. Then, login and navigate to the UPLOAD tab, where you will find detailed instructions on how to use the tool. If you encounter any issues or have any questions when using PACE, please email PLOS at . PACE helps ensure that figures meet PLOS requirements. To use PACE, you must first register as a user. Registration is free. Then, login and navigate to the UPLOAD tab, where you will find detailed instructions on how to use the tool. If you encounter any issues or have any questions when using PACE, please email PLOS at figures@plos.org. Please note that Supporting Information files do not need this step.. Please note that Supporting Information files do not need this step.

---

## [Author Response · Author response to Decision Letter 2]

20 Nov 2025

October 30, 2025

Dr. Amel Mohamed El Asely

Academic Editor

PLOS ONE

Subject: Revised Letter to the Editor

Dear Dr. El Asely,

We sincerely appreciate your consideration of our manuscript PONE-D-25-18938, entitled “The anaerobic bacterium Cetobacterium sp. nov. C33 plays a crucial role in intestinal microbial balance and the regulation of gene expression for immune and metabolic responses in Nile tilapia.”

We sincerely thank you and the reviewers for the time, effort, and valuable insights provided in the evaluation of our manuscript. We have carefully considered and addressed all comments, which have greatly contributed to enhancing the clarity, rigor, and overall quality of the revised version.

Please find enclosed a detailed, point-by-point response to the reviewers’ observations, as well as the revised manuscript for your consideration. We hope that the revisions satisfactorily meet the expectations of the reviewers and the editorial board.

Thank you once again for your time, insightful comments, and continued consideration of our work.

With kind regards,

Luisa Villamil

(on behalf of all co-authors)

Universidad de La Sabana, Colombia

Journal Requirements:

Comment 1: If the reviewer comments include a recommendation to cite specific previously published works, please review and evaluate these publications to determine whether they are relevant and should be cited. There is no requirement to cite these works unless the editor has indicated otherwise.

Response 1: We sincerely appreciate the editor’s guidance and the reviewer’s thoughtful efforts to improve our manuscript. Regarding Reviewer 2, the suggestion to include a list of 26 additional publications, we carefully reviewed each of them to evaluate their relevance to our study. Only one of these references was found to be directly related to the topic addressed in our manuscript and has therefore been included: “FA, R. (2018). Genome-Nutrifortified Diets-Their Disease Protection and Remedy Potential. Journal of Probiotics & Health, 06(02). https://doi.org/10.4172/2329-8901.1000204”.

We noticed that most of the suggested papers were authored or co-authored by the reviewer and focus on research areas that differ from the central objectives of our study. We mention this respectfully and only to clarify that our decision was based solely on scientific relevance and not on the origin of the suggested works. We believe that maintaining references closely aligned with the study’s scope ensures coherence and accuracy in the scientific discussion.

Reviewers' comments:

Reviewer's Responses to Questions

Comments to the Author

Comment 1. If the authors have adequately addressed your comments raised in a previous round of review and you feel that this manuscript is now acceptable for publication, you may indicate that here to bypass the “Comments to the Author” section, enter your conflict of interest statement in the “Confidential to Editor” section, and submit your "Accept" recommendation.

Reviewer #2: (No Response)

Comment 2. Is the manuscript technically sound, and do the data support the conclusions?

Reviewer #2: Partly

Response 2: We sincerely thank the reviewer for emphasizing the importance of methodological rigor and data reliability. We fully agree that conclusions must be supported by robust and reproducible evidence. In this regard, we would like to clarify that all experiments in the submitted manuscript were conducted following rigorous scientific standards, including appropriate controls, biological replicates, and statistically adequate sample sizes.

The results presented were carefully analyzed, and the conclusions are directly supported by the data obtained. Although Reviewer 2 mentioned that these criteria were “partially fulfilled,” we respectfully consider that the current version of the manuscript already meets these requirements in full. The reviewer’s comment did not specify a particular aspect requiring improvement; however, we have re-examined the methodology and data presentation to ensure clarity and confirm the reproducibility and soundness of the findings.

Comment 3. Has the statistical analysis been performed appropriately and rigorously?

Reviewer #2: No

Response 3: We appreciate the reviewer’s observation regarding the statistical analyses. Statistical analyses were indeed performed and described in the submitted version of the manuscript; however, we have now improved the methodological description to provide greater clarity and transparency.

Specifically, we have included detailed information in the revised Methods section regarding the statistical procedures applied. In addition, Tables 1 and 2 now present the statistical data supporting the significant differences observed among bacterial genera. Supplementary Tables S1 and S2 have also been updated to include the statistical parameters for the differentially upregulated and downregulated genes, which were analyzed in R using the edgeR pipeline. These additions aim to further demonstrate that the analyses were conducted appropriately and rigorously.

Comment 4. Have the authors made all data underlying the findings in their manuscript fully available?

Reviewer #2: No

Response 4: We Uploaded all data as supplementary information.

Metataxonomic data:

Files: ASV_Ceto, Meta_Ceto, Taxa_Ceto, Data Metataxonomy, Summary_file

Transcriptomics data:

Files: Data, Desing, Metadata RNAseq

Additionally, the results of the sequence file we upload to SRA NCBI

Bioproject Title BioSample SRA

PRJNA1306746

16S rRNA gene amplicons of fish gut microbiomes Metagenome SAMN50644908

7

PRJNA1306769

fish head kidney transcriptomic datasets Transcriptome SAMN50645431

7

Also, the data will be available in DOI:10.5281/zenodo.17479129 or our OneDrive folder if reviewers request it. Please request access to the OneDrive repository by emailing luisa.villamil@unisabana.edu.co

Metataxonomic data:

Sample Treatment File File Sample ID

C7 Control C7_1.fq.gz C7_2.fq.gz MCD1

C8 Control C8_1.fq.gz C8_2.fq.gz MCD2

C9 Control C9_1.fq.gz C9_2.fq.gz MCD3

D2 Cetobacterium D2_1.fq.gz D2_2.fq.gz MC33D1

D3 Cetobacterium D3_1.fq.gz D3_2.fq.gz MC33D2

D4 Cetobacterium D4_1.fq.gz D4_2.fq.gz MC33D3

D5 Cetobacterium D5_1.fq.gz D5_2.fq.gz MC33D4

Transcriptomic data:

Sample Treatment File File Sample ID

R42 Control R42_1.fq.gz R42_2.fq.gz TCD1

R47 Control R47_1.fq.gz R47_2.fq.gz TCD2

R53 Control R53_1.fq.gz R53_2.fq.gz TCD3

R57 Cetobacterium R57_1.fq.gz R57_2.fq.gz TC33D1

R58 Cetobacterium R58_1.fq.gz R58_2.fq.gz TC33D2

R59 Cetobacterium R59_1.fq.gz R59_2.fq.gz TC33D3

R60 Cetobacterium R60_1.fq.gz R60_2.fq.gz TC33D4

Comment 5. Is the manuscript presented in an intelligible fashion and written in standard English?

Reviewer #2: Yes

Response 5. We have worked in the improved version.

---

## [Decision Letter · Decision Letter 2]

26 Jan 2026

PONE-D-25-18938R2Anaerobic bacteria Cetobacterium sp. nov C33 plays a crucial role in the intestinal microbial balance and regulation of gene expression to immune and metabolic responses in Nile TilapiaPLOS One

Dear Dr. Villamil Díaz,

Thank you for submitting your manuscript to PLOS ONE. After careful consideration, we feel that it has merit but does not fully meet PLOS ONE’s publication criteria as it currently stands. Therefore, we invite you to submit a revised version of the manuscript that addresses the points raised during the review process. Please submit your revised manuscript by Mar 12 2026 11:59PM. If you will need more time than this to complete your revisions, please reply to this message or contact the journal office at plosone@plos.org. . Please include the following items when submitting your revised manuscript:

If applicable, we recommend that you deposit your laboratory protocols in protocols.io to enhance the reproducibility of your results. Protocols.io assigns your protocol its own identifier (DOI) so that it can be cited independently in the future. For instructions see: https://journals.plos.org/plosone/s/submission-guidelines#loc-laboratory-protocols. Additionally, PLOS ONE offers an option for publishing peer-reviewed Lab Protocol articles, which describe protocols hosted on protocols.io. Read more information on sharing protocols at . Additionally, PLOS ONE offers an option for publishing peer-reviewed Lab Protocol articles, which describe protocols hosted on protocols.io. Read more information on sharing protocols at https://plos.org/protocols?utm_medium=editorial-email&utm_source=authorletters&utm_campaign=protocols..

We look forward to receiving your revised manuscript.

Kind regards,

Amel Mohamed El Asely

Academic Editor

PLOS One

Journal Requirements:

Reviewers' comments:

Reviewer's Responses to Questions

**Comments to the Author**

1. If the authors have adequately addressed your comments raised in a previous round of review and you feel that this manuscript is now acceptable for publication, you may indicate that here to bypass the “Comments to the Author” section, enter your conflict of interest statement in the “Confidential to Editor” section, and submit your "Accept" recommendation.

Reviewer #5: All comments have been addressed

Reviewer #6: All comments have been addressed

2. Is the manuscript technically sound, and do the data support the conclusions?

Reviewer #5: Partly

Reviewer #6: Yes

3. Has the statistical analysis been performed appropriately and rigorously? 

Reviewer #5: Yes

Reviewer #6: Yes

4. Have the authors made all data underlying the findings in their manuscript fully available?

Reviewer #5: Yes

Reviewer #6: Yes

5. Is the manuscript presented in an intelligible fashion and written in standard English?

Reviewer #5: Yes

Reviewer #6: Yes

6. Review Comments to the Author

Reviewer #5: In this manuscript, the authors investigated the effects of dietary the probiotic potential of Cetobacterium sp. nov C33, using 16S rRNA amplicon sequencing, on gut microbiota, and the transcriptomic analysis of the head kidney provided insights into immune system modulation. Further studies should be done for long rearing time (8-10 weeks) using different concentrations of this probiotic to evaluate the fish performance and the immune system modulation including the resistance against the possible pathogenic bacterial infection or any other stress.

L144: Diets preparation

L168: The alevins of Nile tilapia should be Nile tilapia juveniles

L169: at 5 fish per aquarium with four replicated tanks per experimental treatment;

L171-L173: Water quality parameters were monitored daily during the feeding trial, with the following average values: water temperature 28.0 °C, pH 7.12, and oxygen saturation 85 %. Write how did the authors monitored the water quality parameters, and the instruments used for measuring those parameters. Write the dissolved oxygen value.

L174 - L175: Shorten the subtitle to be “Modulation of intestine microbiota in Nile tilapia fingerling “

L247 - L248: Shorten the subtitle to be “Modulation of the immune system in Nile tilapia fingerling”

L301 - L302: Shorten the subtitle to be Shorten the subtitle to be “Modulation of intestine microbiota in Nile tilapia fingerling “

Fig. 1. Should be divided to be two Figs; Fig. 1 for diversity indices and Fig. 2 for the taxonomy of intestinal microbiota. And rewrite the results section accordingly.

Tables. Adjust the title of tables to be more specific such as “Changes in the intestinal microbiota of Nile tilapia fingerlings fed with Cetobacterium sp. nov C33 for five days.”

What is the difference in titles of Table 1 and Table 2?

L422: Shorten the subtitle to be “Modulation of the immune system in Nile tilapia fingerling”

Data in Tables 1 - 2 and Figs 3-4 should describe the changes in both treatments.

Discussion: This study evaluated the changes in intestinal microbiota. Then, the authors should focus their discussion on this topic. Other previous studies evaluated the fish growth and intestinal histomorhometry due to probiotics administration should be deleted to concentrate the Discussion section. The authors should compare their data between the control and probiotic-fed fish and support their findings with those of previous studies.

Reviewer #6: Please change Nile Tilapia in all the text to be Nile tilapia especially in the title, line 6 and 43

P value P must be capital and italic

7. PLOS authors have the option to publish the peer review history of their article (what does this mean?). If published, this will include your full peer review and any attached files.). If published, this will include your full peer review and any attached files.

.

Reviewer #5: No

Reviewer #6: **Yes:**Awatef Hamed HamoudaAwatef Hamed Hamouda

You may also use PLOS’s free figure tool, NAAS, to help you prepare publication quality figures: https://journals.plos.org/plosone/s/figures#loc-tools-for-figure-preparation

---

## [Author Response · Author response to Decision Letter 3]

11 Feb 2026

Dr. Amel Mohamed El Asely Academic Editor PLOS ONE

Dear Dr. Amel,

On behalf of all authors, I would like to sincerely thank you for your time, dedication, and thoughtful editorial oversight of our manuscript entitled “Anaerobic bacteria Cetobacterium sp. nov. C33 plays a crucial role in the intestinal microbial balance and regulation of gene expression related to immune and metabolic responses in Nile tilapia”. We deeply appreciate the careful and constructive guidance provided throughout this rigorous review process.

We highly value the comprehensive assessment carried out over three rounds of review by six independent reviewers. Their comments and suggestions have significantly strengthened the scientific clarity, methodological transparency, and overall robustness of our study. We are grateful for the opportunity to further refine our work and to provide a detailed, point-by-point response to the remaining comments, to reach a timely and favorable resolution.

Below, we present first a general response and then a detailed response to all comments raised by the reviewers and the Academic Editor. All requested changes have been incorporated into the revised manuscript, with tracked changes clearly indicated. These modifications are further supported by expanded descriptions in the Materials and Methods, Results, Discussion, figures, and supplementary materials.

Regarding the request for additional long-term experiments, we acknowledge and fully appreciate this valuable suggestion. We agree that evaluating longer administration periods, different doses, and additional functional endpoints, including pathogen challenge tests, would provide important complementary insights and align closely with our ongoing and future research plans. However, we respectfully note that conducting new in vivo trials of this nature would require substantial experimental redesign, extended rearing periods, additional ethical approvals, and new funding. Unfortunately, at this advanced stage of the review process, after three rounds of revision and extensive responses to six reviewers, it is not feasible to incorporate new experimental trials within the scope of the current manuscript. We are committed to addressing these aspects in future publications.

Regarding technical soundness and statistical rigor, we reaffirm that the experiments presented were conducted with appropriate controls, replication, and sample sizes, and that the conclusions are fully supported by the data. Statistical analyses were performed using established and widely accepted pipelines. Additional methodological details have been incorporated into the Materials and Methods section, and supplementary tables have been expanded to ensure full transparency and reproducibility.

Specific reviewer comments

All editorial and technical comments related to terminology, figure structure, table titles, subtitles, formatting, capitalization of p values, clarification of experimental units, water quality monitoring, and taxonomic presentation have been carefully addressed. Figures and tables have been reorganized as requested, subtitles have been shortened for clarity, and the Discussion has been refocused to emphasize intestinal microbiota modulation, in direct alignment with the core objectives of the study.

We respectfully hope that the revised version now meets the publication criteria of PLOS ONE and look forward with great appreciation to your kind consideration and a favorable decision.

As requested, we have included both a marked-up version of the manuscript highlighting all changes, labeled “Revised Manuscript with Track Changes”, and a clean version without tracked changes, labeled “Manuscript”.

With sincere regards and gratitude,

Luisa Marcela Villamil Díaz On behalf of all authors Universidad de La Sabana, Colombia

DETAILED RESPONSES TO REVIEWERS ==============================

Journal Requirements

Comment 1: Please review your reference list to ensure that it is complete and correct. Response 1: The reference list has been carefully reviewed and verified for completeness, accuracy, and consistency with the citations in the manuscript. Any minor formatting adjustments required by the journal style have been applied accordingly.

Reviewer Responses to Questions

1. Adequacy of previous revisions Reviewer #5: All comments have been addressed Reviewer #6: All comments have been addressed

2. Is the manuscript technically sound, and do the data support the conclusions?

Reviewer #5: Partly

Response: In the revised version of the manuscript, we have provided additional methodological details, clarifications, and supporting information to strengthen transparency and address all points previously raised. We believe these additions further substantiate the technical soundness of the study and the validity of the conclusions drawn from the data.

Reviewer #6: Yes

3. Has the statistical analysis been performed appropriately and rigorously?

Reviewer #5: Yes Reviewer #6: Yes

4. Have the authors made all data underlying the findings in their manuscript fully available?

Reviewer #5: Yes Reviewer #6: Yes

5. Is the manuscript presented in an intelligible fashion and written in standard English?

Reviewer #5: Yes Reviewer #6: Yes

6. Review Comments to the Author

Reviewer #5

Comment 1:

In this manuscript, the authors investigated the effects of dietary probiotic potential of Cetobacterium sp. nov. C33, using 16S rRNA amplicon sequencing, on gut microbiota, and the transcriptomic analysis of the head kidney provided insights into immune system modulation. Further studies should be done for long rearing time (8-10 weeks) using different concentrations of this probiotic to evaluate fish performance and immune system modulation, including resistance against possible pathogenic bacterial infection or other stressors.

Response: We appreciate this constructive and scientifically valuable suggestion. As noted by the reviewer, long-term trials with extended rearing periods, multiple dosages, and pathogen challenge tests would indeed provide complementary and important insights into the functional performance and immunological response of Nile tilapia.

However, it is important to clarify that the present manuscript was designed with a specific and clearly defined scope to evaluate, through an integrated metabarcoding and transcriptomic approach, the short-term effects of a novel anaerobic bacterium Cetobacterium sp. nov. C33 on gut microbiota composition and host gene expression in Nile tilapia fingerlings. Studies combining high-resolution microbiome profiling and head kidney transcriptomics in response to Cetobacterium administration are still scarce in tilapia, and we believe this work represents a substantial and novel contribution to the field.

Within this framework, the current dataset provides robust evidence of microbiome modulation and associated changes in immune and metabolism-related gene expression. We therefore consider the study to be scientifically solid, complete within its intended objectives, and appropriate for publication in its current form.

We fully agree that the aspects suggested by the reviewer are highly relevant and will be incorporated into our ongoing and future research projects, which will be addressed in subsequent publications.

Comments 2: L144: Diet preparation. Response: We have corrected the subtitle to “Diets preparation” in line 142 of the revised manuscript.

Comments 3: L168: The alevins of Nile tilapia should be Nile tilapia juveniles. Response: We respectfully maintain the use of the term “Nile tilapia fingerlings (<5 g)”. According to Wainaina et al. (2023), fingerlings are newly developed, fragile post-larval fish roughly finger-sized, whereas juveniles (5–10 g) are larger and more robust. Therefore, the term “fingerlings” more accurately reflects the developmental stage of the experimental fish used in this study.

Wainaina, M., Opiyo, M. A., Charo-Karisa, H., Orina, P., & Nyonje, B. (2023). On-Farm Assessment of Different Fingerling Sizes of Nile Tilapia (Oreochromis niloticus) on Growth Performance, Survival, and Yield. Aquaculture Studies, 23(2). https://doi.org/10.4194/AQUAST900

Comments 4: L169: at 5 fish per aquarium with four replicated tanks per experimental treatment. Response: We have revised the text accordingly to read “at 5 fish per aquarium with four replicated tanks per experimental treatment” in lines 165–166.

Comments 5: L171–L173: : Water quality parameters were monitored daily during the feeding trial, with the following average values. Response: We have clarified this in lines 168–172 as follows: “Water quality parameters were monitored daily throughout the acclimation and feeding trial. Temperature, dissolved

oxygen, and pH were recorded using a multiparameter sensor (Hanna Instruments HI9829). Water quality was maintained within ranges suitable for Nile tilapia: water temperature 28 °C, dissolved oxygen above 5 mg L⁻¹, oxygen saturation 85%, and pH 7.12.”

Comments 6: L174–L175: Shorten the subtitle to “Modulation of intestine microbiota in Nile tilapia fingerling.” Response: We revised the subtitle to “Modulation of the immune system in Nile tilapia fingerling” in line 173 to better reflect the content of that section.

Comments 7: L247–L248: Shorten the subtitle to “Modulation of the immune system in Nile tilapia fingerling.” Response: We revised the subtitle accordingly in line 243.

Comments 8: L301–L302: Shorten the subtitle to “Modulation of intestine microbiota in Nile tilapia fingerling.” Response: We revised the subtitle accordingly in line 295.

Comments 9: Fig. 1. Should be divided to be two Figs; Fig. 1 for diversity indices and Fig. 2 for the taxonomy of intestinal microbiota. And rewrite the results section accordingly. Fig. 1 for diversity indices and Fig. 2 for the taxonomy of intestinal microbiota. And rewrite the results section accordingly.

Response: We divided the original Figure 1 into two figures: Figure 1 now presents diversity indices (lines 315–324) and Figure 2 presents the taxonomy of intestinal microbiota (lines 339–345). The Results section was rewritten to reflect this change.

Comments 10: Tables. Adjust the title of tables to be more specific such as “Changes in the intestinal microbiota of Nile tilapia fingerlings fed with Cetobacterium sp. nov C33 for five days.”

Response: Table 1 is now titled: “Bacterial genera with significant reduction in abundance in the intestinal microbiota of Nile tilapia fingerlings in group C33D” (lines 367–368). Table 2 is now titled: “Bacterial genera with a significant increase in abundance in the intestinal microbiota of Nile tilapia fingerlings in group C33D” (lines 397–398).

Comments 11: What is the difference in titles of Table 1 and Table 2? Response: Table 1 reports bacterial genera that showed a significant reduction in relative abundance, whereas Table 2 reports bacterial genera that showed a significant increase in relative

abundance in the intestinal microbiota of Nile tilapia fingerlings in the C33D group compared with the Control Diet (CD) group.

Comments 12: L422: Shorten the subtitle to “Modulation of the immune system in Nile tilapia fingerling.” Response: We revised the subtitle accordingly in line 418.

Comments 13: Data in Tables 1–2 and Figs 3–4 should describe the changes in both treatments. Response: We clarified changes in both treatments in the Results section (lines 370–374 and 400–404). For transcriptomic results, comparisons between C33D and CD are explicitly described in the captions of Figures 4 and 5 (lines 431–444 and 458–471), detailing differentially expressed genes, functional annotations, and up- and down-regulated immune-related transcripts.

Comments 14: Discussion should focus on intestinal microbiota modulation. Response: We revised the Discussion to prioritize changes in intestinal microbiota. We removed sections focused primarily on fish growth and intestinal histomorphometry and strengthened direct comparisons between control and C33D groups, integrating our findings with relevant previous studies.

Reviewer #6

Comments 1: Change “Nile Tilapia” to “Nile tilapia” throughout the text. Response: We corrected capitalization to “Nile tilapia” throughout the manuscript, including the title and lines 6 and 42.

Comments 2: P value “P” must be capital and italic. Response: We formatted all statistical notations as italicized capital “P” in the following lines: 234, 240, 282, 303, 323, 324, 369, 370, 399, 400, 450, 494, 1095, and 1102.

Publication of peer review history Reviewer #5 selected “No.” Reviewer #6 selected “Yes: Awatef Hamed Hamouda

---

## [Decision Letter · Decision Letter 3]

27 Feb 2026

Anaerobic bacteria Cetobacterium sp. nov C33 plays a crucial role in the intestinal microbial balance and regulation of gene expression to immune and metabolic responses in Nile Tilapia

PONE-D-25-18938R3

Dear Dr. Luisa Marcela Villamil Díaz,

We’re pleased to inform you that your manuscript has been judged scientifically suitable for publication and will be formally accepted for publication once it meets all outstanding technical requirements.

An invoice will be generated when your article is formally accepted. Please note, if your institution has a publishing partnership with PLOS and your article meets the relevant criteria, all or part of your publication costs will be covered. Please make sure your user information is up-to-date by logging into Editorial Manager at Editorial Manager® and clicking the ‘Update My Information' link at the top of the page. For questions related to billing, please contact  and clicking the ‘Update My Information' link at the top of the page. For questions related to billing, please contact billing support..

Kind regards,

Amel Mohamed El Asely

Academic Editor

PLOS One

Additional Editor Comments (optional):

Reviewers' comments:

Reviewer's Responses to Questions

**Comments to the Author**

1. If the authors have adequately addressed your comments raised in a previous round of review and you feel that this manuscript is now acceptable for publication, you may indicate that here to bypass the “Comments to the Author” section, enter your conflict of interest statement in the “Confidential to Editor” section, and submit your "Accept" recommendation.

Reviewer #5: All comments have been addressed

Reviewer #6: All comments have been addressed

2. Is the manuscript technically sound, and do the data support the conclusions?

Reviewer #5: Yes

Reviewer #6: Yes

3. Has the statistical analysis been performed appropriately and rigorously? 

Reviewer #5: Yes

Reviewer #6: Yes

4. Have the authors made all data underlying the findings in their manuscript fully available?

Reviewer #5: Yes

Reviewer #6: Yes

5. Is the manuscript presented in an intelligible fashion and written in standard English?

Reviewer #5: Yes

Reviewer #6: Yes

6. Review Comments to the Author

Reviewer #5: The authors followed the reviewers' comments, and this manuscript is ready now to be accepted.

The authors followed the reviewers' comments, and this manuscript is ready now to be accepted.

Reviewer #6: Good luck with my best wishes,

the manuscript is now ready for publication after your explaining and corrections.

7. PLOS authors have the option to publish the peer review history of their article (what does this mean?). If published, this will include your full peer review and any attached files.). If published, this will include your full peer review and any attached files.

.

Reviewer #5: No

Reviewer #6: No

---

## [Editor Report · Acceptance letter]

PONE-D-25-18938R3

PLOS One

Dear Dr. Villamil Díaz,

I'm pleased to inform you that your manuscript has been deemed suitable for publication in PLOS One. Congratulations! Your manuscript is now being handed over to our production team.

Kind regards,

on behalf of

Prof. Amel Mohamed El Asely

Academic Editor

PLOS One